

# Boundary chaos: Exact entanglement dynamics

Felix Fritzsch[1⋆], Roopayan Ghosh[1,2] and Tomaž Prosen[1]

**1** Physics Department, Faculty of Mathematics and Physics,
University of Ljubljana, Ljubljana, Slovenia
**2** Department of Physics and Astronomy, University College London,
Gower Street, London, United Kingdom, WC1E6BT

⋆ felix.fritzsch@fmf.uni-lj.si

## Abstract

We compute the dynamics of entanglement in the minimal setup producing ergodic and mixing quantum many-body dynamics, which we previously dubbed *boundary chaos*. This consists of a free, non-interacting brickwork quantum circuit, in which chaos and ergodicity is induced by an impurity interaction, i.e., an entangling two-qudit gate, placed at the system's boundary. We compute both the conventional bipartite entanglement entropy with respect to a connected subsystem including the impurity interaction for initial product states as well as the so-called operator entanglement entropy of initial local operators. Thereby we provide exact results in a particular scaling limit of both time and system size going to infinity for either very small or very large subsystems. We show that different classes of impurity interactions lead to very distinct entanglement dynamics. For impurity gates preserving a local product state forming the bulk of the initial state, entanglement entropies of states show persistent spikes with period set by the system size and suppressed entanglement in between, contrary to the expected linear growth in ergodic systems. We observe similar dynamics of operator entanglement for generic impurities. In contrast, for T-dual impurities, which remain unitary under partial transposition, we find entanglement entropies of both states and operators to grow linearly in time with the maximum possible speed allowed by the geometry of the system. The intensive nature of interactions in all cases causes entanglement to grow on extensive time scales proportional to system size.

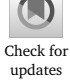

# 1 Introduction

One of the hallmark features of quantum mechanics which has no counterpart in classical physics is quantum entanglement. With the advent of quantum computing and communication, quantum entanglement has become an important resource to overcome limitations of classical computing [1]. Futhermore, the study of the creation and kinetics of entanglement is currently a well established tool to characterize the complex dynamics of condensed matter systems and the emergence of their thermodynamic description both theoretically [2,3] and experimentally [4–7]. Simultaneously the creation of the so-called entanglement entropy sets fundamental limits on how such systems can be simulated on classical computers [8–11].

A standard protocol to investigate the creation of entanglement in a quantum system is that of a quench, i.e., the time evolution from an initial state with typically little or no entanglement, which is not an eigenstate of the system's evolution operator. A very general feature in such a non-equilibrium situation is the linear growth of entanglement between disjoint subsystems measured by, for example, the von Neumann or Rényi entropies of the reduced density matrix, and their subsequent saturation for finite systems.

This linear growth has been observed in distinguished scenarios, including experimental setups with cold atoms [5], and has been explained by different mechanisms. In integrable systems, the linear growth can be traced back to propagating stable quasi particles [12–18]. But it has also been shown that the linear growth is ubiquitous even in the absence of stable quasi particle excitations [19–30]. In spatially local chaotic many-body systems it can be qualitatively deduced from a minimal membrane separating the subsystems [23,24]. Recently, more rigorous results on the nature of the linear growth of entanglement in chaotic systems

have been obtained for random quantum circuits [23, 25, 26] or Floquet systems including, e.g., periodically driven chaotic spin chains [27] or Floquet quantum circuits [28–30]. In particular in the latter a dual description of the dynamics under a swapping of space and time allows for rigorous results in the thermodynamic limit [16, 27, 30–32].

Nevertheless, there are some notable exceptions from the linear growth of entanglement entropies, e.g., in the presence of confined quasiparticles [33] as well as in disordered [34] and many-body localized systems [35–39], which display logarithmic growth.

An alternative point of view for characterizing the dynamics of many-body systems is provided by the growth of complexity of initially simple operators, for example, local operators, under Heisenberg time evolution. There are various ways to characterize the complexity of the time evolved operator, including out-of-time-ordered correlators, which quantify the scrambling of operators and the growth of their support [25, 40–46] as well as their Krylov complexity [47–50]. Moreover, one can study correlations in the time evolved operator shared by disjoint subsystems.

By interpreting operators as states in an enlarged Hilbert space by means of an operator to state mapping, this idea can be made concrete by applying the concept of entanglement to the vectorized operator. This leads to the notion of operator entanglement, which originally was introduced to study the entanglement properties of evolution operators or, more generally, quantum channels [51]. In the context of many-body systems this measure can also be used to quantify the growth of complexity of initially simple operators [52, 53]. The latter provides additional insight into the complexity of the many body dynamics and sets limits for the numerical simulation of Heisenberg time evolution of operators [52–56].

Previously, the aforesaid quantity has been studied in various settings, including systems with local solitons [57] as well as integrable systems [52, 53, 56, 58, 59] and conformal field theories [60], where logarithmic growth of operator entanglement entropies were observed. In contrast, in general chaotic systems entanglement entropies initially grow linear in time [24, 61, 62] until they eventually saturate. An interesting exception from saturation at late times, the so-called entanglement barrier, occurs for the reduced density matrix of a pure state in systems with short range interactions [60, 63, 64]. There, after initially growing, entanglement entropies ultimately drop down again until they settle at the value of a lowly entangled thermal state.

In this paper we consider both the entanglement dynamics for product states and the operator entanglement dynamics for local operators in a simple quantum circuit setting, which allows for exact solutions in the limit of large system size $L \to \infty$. More precisely we study a free quantum circuit model, with trivial free dynamics, which we perturb at the system's boundary with an entangling two-qudit gate, which we call an *impurity interaction*. This setup was introduced in Ref. [65] and dubbed *boundary chaos*. One might think of such a circuit as a toy model for a free system subject to a local perturbation which introduces nontrivial scattering to the otherwise free dynamics. Despite its simple nature the boundary chaos circuit has been shown to be quantum chaotic in the sense of spectral statistics and exhibits ergodic dynamics [65].

Moreover, this setting allows us to analytically integrate out the free part of the dynamics, in a conceptually similar fashion as for Poincaré maps in classical dynamics [66]. This enables us to provide a simplified tensor network representation of the time evolved reduced density matrix, or super density matrix in the case of operator entanglement. In particular, these networks contain only the impurity interaction and hence contain a factor of $1/L$ less gates. This renders them amenable for efficient numerical contraction in terms of suitable transfer matrices even for very large systems, and for analytical calculations in the thermodynamic limit of system size $L \to \infty$. Depending on the choice of impurity interaction we either obtain the reduced (super) density matrix or the corresponding Rényi entanglement entropies exactly. For

different classes of impurity interactions we find very different entanglement dynamics including exponentially suppressed (operator) entanglement entropies for most times accompanied by periodical spikes as well as maximally fast linear growth of (operator) entanglement. In all cases, however, we find entanglement to grow on extensive time scales $\sim L$. We summarize our results in more detail in the following section.

## 1.1 Summary of results

To discuss our results, let us first introduce some notation. The free part of the boundary chaos circuit is built from swap gates on a chain of qudits, i.e., $q$-level systems of length $L + 1$. Chaos and ergodicity is introduced by placing an impurity interaction, i.e., a non-trivial two-qudit gate $U$ just at the system boundary. *Remarkably, this is indeed enough to make the system ergodic!* Namely, spectral fluctuations of the evolution operator coinicide with those of appropriate random matrix ensembles, see App. A, and dynamical correlations decay exponentially in time [65]. In this work we use the simplistic nature of this model to obtain the asymptotics of entanglement dynamics analytically for different classes of impurity interactions, yielding results that seem quite non-intuitive at first glance. We systematically compute the entanglement dynamics of initial product states and of local operators, which provides insight to the complex many-body dynamics both for our simple model as well as for generic lattice systems. To define entanglement entropies we introduce a bipartition of the system into a subset $A$ containing the first $l + 1$ qudits, and its complement $\bar{A}$, containing the remaining $L - l$ qudits. For this bipartion we compute the dynamics of the reduced (super) density matrix $\rho_l(t)$, and its Rényi entropies $R_n = \ln \text{tr}(\rho_l(t)^n)/(1 - n)$, for initial product states and local operators in the scaling limit $L, t \to \infty$ with $t/L$ fixed. Depending on the type of impurity interaction we obtain exact results either for fixed subsystem size $l$ or in the limit $l \to \infty$. We describe the different classes of impurity interactions and the results obtained for them below.

1. *Product initial state and impurity interactions with a vacuum state:* These interactions preserve a certain 2-qudit product state[1] $|\circ\circ\rangle$. For example, for spin qubits we can take this to be the state with both spins pointing up ($|\circ\rangle = |\uparrow\rangle$). Starting from initial states of the form $|\bullet \circ \circ \cdots \circ \circ\rangle$ with a single localized excitation $|\bullet\rangle$ (e.g. $|\downarrow\rangle$) at the boundary, for finite systems, we see persistent revivals of $R_n$ with period given by the system size $L$, see Fig. 1 below.

   This seemingly contradicts the ergodic-like spectral statistics of the evolution operator, as in such a case a monotonic growth of entanglement is expected. However, because our model has the impurity just at the boundary, the initially localized excitation travels completely into the - typically much larger - complement of the considered subsystem and leaves only the vacuum in the subsystem close to the boundary. Only at resonant times, i.e., integer multiples of $L$, has the excitation traveled ballistically through the system and returned to the impurity. And this is the only time where correlations between the subsystem and its complement might develop. Thus, exclusively at the boundary, the excitations can scatter into higher excited states which lead to a growth of entanglement. As this process occurs on a time scale proportional to $L$ entanglement can grow at most on extensive time scales (time proportional to $L$).

   To put it more concretely, we find the reduced density matrix for large system size to be close to the pure state

$$\rho_l(t) = |\circ \circ \cdots \circ \circ\rangle\langle\circ \circ \cdots \circ \circ| \,, \tag{1}$$

---

[1] A trivial example is a U(1)-symmetric, i.e. particle number conserving interaction gate. Here, however, we consider more general gates which involve also the transitions $|\bullet\circ\rangle \to |\bullet\bullet\rangle$, $|\circ\bullet\rangle \to |\bullet\bullet\rangle$.

for most times. For finite systems, introducing $\tau = \lfloor t/L \rfloor$ and a remainder $\delta = t \bmod L$ such that $t = \tau L + \delta$, we observe that Eq. (1) holds up to terms exponentially suppressed as $\lambda_0^\delta$ for some $\lambda_0 < 1$, controlled by a subleading eigenvalue of certain transfer matrices. As the Rényi entropy of a pure state vanishes it is dominated by the subleading term and reads

$$R_n(t) \sim |\lambda_0|^\delta . \tag{2}$$

This indicates exponential suppression of Rényi entropies with $\delta$ as well for most times as $L \to \infty$. For resonant times, $t \approx \tau L$ , the entanglement entropy is of order one.

2. *Product initial state and T-dual impurity interactions:* T-dual gates are those two-qudit gates which remain unitary under partial transposition (on a single qudit). Even though the asymptotic reduced density matrix can not be obtained explicitly, the corresponding Rényi entropies can be computed exactly for large subsystem size $l \to \infty$. In contrast to the previous case we recover the result expected for fully chaotic systems given by

$$R_n(t) = 2\tau \ln(q) + \text{const.} , \tag{3}$$

independent from the Rényi index $n$, implying flat entanglement spectrum of $\sim e^{R_n(t)}$ non-zero eigenvalues of $\rho_l(t)$. Noting that $\tau \sim t/L$ the above equation describes linear growth of $R_n$ with time at maximum velocity $\ln(q)/L$ allowed by the system's geometry. The only difference to a spatially homogeneous chaotic system is the additional $1/L$ factor, which is due to the density $1/L$ of nontrivial interactions. Moreover, Eq. (3) suggests a staircase structure of the entanglement entropies $R_n(t)$ with steps at integer values of $t/L$. Such staircaise structure, but with different step height, is also observed for finite subsystems until $R_n(t)$ saturates at late times, see Fig 2. The saturation value for finite systems, however, depends on the impurity interaction at the boundary.

3. *Product initial state and generic impurity interactions:* In this case we see a mixture of the above two scenarios. This is depicted in Fig. 3. The leading eigenvalue is still 1, which leads to some plateau of $R_n$ for fixed $\tau$ independent of $L$. However, unlike in the T-dual case, there are subleading transfer matrix eigenvalues $\lambda_0$ which lead to additional structure $\sim \lambda_0^\delta$ on top of the plateau.

4. *Local operator and generic impurity interactions:* The concept of entanglement and entanglement entropies can also be applied to vectorized operators, i.e. using an operator-to-state mapping, where the operators are interpreted as states in an enlarged Hilbert space. For generic impurity interactions the vectorized identity operator $|\circ\rangle = |\mathbb{1}_q\rangle$ plays the role of a vacuum state as a consequence of unitality of Heisenberg time evolution similar to the case of states and vacuum-preserving impurity interactions. For local operators the role of the excitation is now played by the nontrivial component of the operator. As a consequence of this analogy the corresponding operator entanglement/Rényi entropies show qualitatively similar dynamics, see Fig. 4 below, and the physical intuition remains the same. The reduced super density matrix is given by the operator version of Eq. (1) for most times and corresponds to a pure state. The latter is just the vectorization of the identity operator $\mathbb{1}_A$ on the subsystem $A$.

5. *Local operator and T-dual impurity interactions:* As was the case for states, the situation changes drastically, if we additionally demand T-duality of the impurity interaction. Using similar arguments as for states we obtain the Rényi entropies for large system and subsystem size exactly. Again we find linear growth of operator entanglement entropies

with time at maximum speed as

$$R_n(t) = 2\tau \ln\left(q^2\right) + \text{const.}, \tag{4}$$

with the only difference to Eq. (3) being the local Hilbert space dimension $q^2$ instead of $q$. For finite subsystems, we again find a similar staircase structure as in the case of states, see Fig. 5.

We would like to reiterate that for both vacuum-preserving and T-dual impurity interactions, spectral fluctuations of the full circuit show similar properties, e.g. level repulsion, rendering the systems quantum chaotic in the sense of spectral statistics. Dynamics of entanglement, in contrast, is strikingly different being either exponentially suppressed for most times in the case of impurities with a vacuum state, whereas it shows linear growth with maximum speed in the T-dual case.

Additionally, while we state the results for initial states where the localized excitation is placed at the edge of the system (where interaction operates), they are qualitatively valid for the excitation placed anywhere in the lattice, in most cases. This can be shown easily for all the cases above, except for the operator entanglement with T-dual gates where the computation is complicated and a simple conclusion cannot be drawn.

In what follows, we shall first explain the setting of the problem and the notations used throughout the rest of the work in Sec. 2. Then, we will derive the tensor network representation of the reduced density matrix in Sec. 3.1; followed by details of the computation of entanglement dynamics of product states in Sec. 3.2 and operator entanglement in Sec. 4. Finally, we conclude by discussing implications of our results and possible extensions in Sec. 5. Moreover, in App. A we provide additional insight into the spectral fluctuations in the boundary chaos circuit and in App B we comment on the subleading part of the transfer matrices' spectra.

## 2 Setting

In this section, we introduce the class of quantum circuits we use to obtain our results. As we shall describe in Sec. 2.1, interactions are introduced only on the boundary and hence we call this a *boundary chaos circuit*. We shall define and briefly discuss the entanglement entropies both for states and operators in Sec. 2.2.

### 2.1 Boundary chaos circuit

We start from the Floquet system generated by a free brickwork quantum circuit on a one dimensional lattice of size $L + 1$, with sites labelled by $i \in \{0, 1, \ldots L\}$. Then, we render the evolution non-trivial by adding a two site non-trivial gate acting on sites 0 and 1. Each site is occupied by a qudit with local Hilbert space given by $\mathbb{C}^q$ having canonical (computational) basis $|\alpha\rangle$ with $\alpha \in \{0, 1, \ldots, q-1\}$. Hence the total Hilbert space $\mathcal{H} = (\mathbb{C}^q)^{\otimes L+1}$ is of dimension $N = q^{L+1}$ and the product basis is denoted by $|\alpha_0 \alpha_1 \cdots \alpha_L\rangle$. There are two types of local 2-qudit gates, the Swap gate $P$ governing the free evolution and the entangling unitary gate $U \in U(q^2)$, the impurity interaction at the boundary. For the brickwork circuit design the

Floquet operator is given by, $\mathcal{U} = \mathcal{U}_2\mathcal{U}_1 \in \mathrm{U}\left(\mathbb{C}^N\right)$ with

$$\mathcal{U}_1 = \prod_{i=1}^{\lfloor L/2 \rfloor} P_{2i-1,2i}\,, \tag{5}$$

$$\mathcal{U}_2 = U_{0,1} \prod_{i=1}^{\lfloor (L-1)/2 \rfloor} P_{2i,2i+1}\,, \tag{6}$$

where $G_{i,j}$ denotes the unitary gate acting as the 2-qudit gate $G = U, P$ at sites $i,j$ and trivially otherwise. Diagrammatically, the circuit can be represented as

$$\mathcal{U} = \qquad \tag{7}$$

with its elementary building blocks

$$P = \qquad \text{and} \quad U = \qquad . \tag{8}$$

Here, the wedges indicate the orientation of the impurity interaction and wires carry the $q$-dimensional Hilbert space $\mathbb{C}^q$.

To compute time evolution of operators in the Heisenberg picture we use a folded picture which introduces a super circuit with larger local Hilbert space dimensions $q^2$ [67, 68]. To this end we define the vectorization of an operator by the isomorphism $\mathrm{End}\left(\mathbb{C}^q\right) \simeq \mathbb{C}^{q^2}$ defined via bilinear extension of

$$\mathrm{End}\left(\mathbb{C}^q\right) \ni |\alpha\rangle\langle\beta| \mapsto |\alpha\rangle \otimes |\beta\rangle \in \mathbb{C}^{q^2}\,. \tag{9}$$

This extends to a vectorization mapping via tensor multiplication $\mathrm{End}\left(\mathbb{C}^N\right) \simeq \mathbb{C}^{N^2}$. Also note that this mapping is unitary with respect to the Hilbert-Schmidt inner product in $\mathrm{End}\left(\mathbb{C}^N\right)$, $\langle A|B\rangle = \mathrm{tr}\left(A^\dagger B\right)$ and the standard inner product in $\mathbb{C}^{N^2}$. Abusing the notation a bit, we also choose an orthonormal basis in the space of vectorized operators, $|\alpha\rangle$ with $\alpha \in \{0, 1, \dots q^2 - 1\}$ in $\mathbb{C}^{q^2}$ where $|0\rangle$ is the vectorization of $\mathbb{1}_q / \sqrt{q} \in \mathrm{End}\left(\mathbb{C}^q\right)$ and $|\alpha\rangle$ is the vectorization of a Hilbert-Schmidt normalized, Hermitian operator, which is orthogonal to the identity and hence traceless. Under this mapping, we can cast the Heisenberg time evolution of operators $A(t) = \mathcal{U}^{-t}A\mathcal{U}^t$ in a quantum circuit formulation. This super circuit is built from folded gates $S = P \otimes P$ and $W = U^{\mathrm{T}} \otimes U^\dagger \in \mathrm{U}(q^4)$. The circuit $\mathcal{W}$ is of the same form as $\mathcal{U}$ but with the two layers interchanged, i.e., $\mathcal{W} = \mathcal{W}_2\mathcal{W}_1$ with

$$\mathcal{W}_1 = W_{0,1} \prod_{i=1}^{\lfloor (L-1)/2 \rfloor} S_{2i,2i+1}\,, \tag{10}$$

$$\mathcal{W}_2 = \prod_{i=1}^{\lfloor L/2 \rfloor} S_{2i-1,2i}\,, \tag{11}$$

where again, $S_{i,j}$ denotes the unitary gate acting nontrivially as the folded swap gate $P$ and, $W_{i,j}$ the folded impurity interaction $U$ acting on sites $i,j$, and trivially otherwise. Diagrammatically, this can be represented as

$$\mathcal{W} = \qquad \tag{12}$$

where

$$S = \diagup\!\!\!\!\diagdown \quad \text{and} \quad W = \blacksquare \,, \tag{13}$$

with the wedge indicating the orientation of the impurity interaction. Due to folding, the wires now carry the $q^2$-dimensional Hilbert space $\mathbb{C}^{q^2}$. It is also worth noting that no matter how one 'folds', the two layers of the original circuit $\mathcal{U}$ and the super circuit $\mathcal{W}$ are always interchanged, which subsequently will lead to subtle differences in the computation of entanglement entropies.

## 2.2 Entanglement entropies

In this section, we provide a short introduction to the entanglement entropies we compute in our work. We begin with the more familiar case of entanglement of states. As mentioned before, we bipartition the system into the subsystem $A = \{0, 1, \dots l\}$ and its complement $\overline{A} = \{l + 1, \dots, L\}$ for $l < L$. The reduced density matrix of the subsystem $A$, denoted by $\rho_l(t)$, at any instant of time is given by,

$$\rho_l(t) = \operatorname{tr}_{\overline{A}}(|\psi(t)\rangle\langle\psi(t)|) \,, \tag{14}$$

where $|\psi(t)\rangle = \mathcal{U}^t |\psi\rangle$ is the time evolved pure initial state $|\psi\rangle \in \mathbb{C}^N$ and the partial trace is taken over the Hilbert space associated with $\overline{A}$. We compute the $n$-th Rényi entropy $R_n$ (for integer $n$) of the reduced density matrix as,

$$R_n(t) = \frac{1}{1-n} \ln\left[\operatorname{tr}(\rho_l(t)^n)\right] \,. \tag{15}$$

The single replica limit $n \to 1$ gives the von-Neumann entropy $R_1(t)$. Note that for a pure state, we have $R_n(t) = 0$. In contrast the fully mixed state, $\rho_l(t) = \mathbb{1}/q^{l+1}$, has a flat entanglement spectrum and gives $R_n(t) = (l+1)\ln(q)$ independent of $n$. In this work we focus on product initial states $|\psi\rangle = \bigotimes_{i=0}^{L} |\psi_i\rangle$ with $|\psi\rangle_i \in \mathbb{C}^q$. We will also restrict the discussion to states with $|\psi_i\rangle = |\psi_j\rangle$ for $i, j > 0$ for simplicity. Nevertheless, depending on the type of impurity interaction our approach might be applicable to arbitrary product states as well.

For entanglement of operators, the above definitions remain the same if the operators are viewed as vectors in an enlarged Hilbert space as it is suggested by the vectorization mapping. Hence, the reduced super density matrix for initial operator $\mathcal{O} \in \operatorname{End}(\mathbb{C}^N)$ with vectorization $|\mathcal{O}\rangle$ is

$$\hat{\rho}_l(t) = \operatorname{tr}_{\overline{A}}(\mathcal{W}^t |\mathcal{O}\rangle\langle\mathcal{O}| \mathcal{W}^{-t}) \,. \tag{16}$$

Similarly as before, for pure states (pure vectorized operators), the entanglement entropies are zero, while for the fully mixed super density matrix one gets $R_n(t) = (l+1)\ln(q^2)$. The only difference is rooted in the local Hilbert space dimensions $q$ vs $q^2$. Note that an analogous notion of operator entanglement can be applied to the evolution operator $\mathcal{U}$ itself or even more general quantum evolutions as well [51], which, e.g., characterizes their entangling power [69]. In this work, however, we will focus on local operator entanglement [52, 53, 62]. That is we consider local operators of the form $a_0 = a \otimes \mathbb{1}_q^{\otimes L}$ acting non-trivially as $a \in \operatorname{End}(\mathbb{C}^q)$ only on the first lattice site. We note that for operators and generic gates as well as for states with vacuum-preserving gates the results are independent from where we put the non-trivial operator/excited state (up to a shift in time). For states and T-dual gates we can go further and perform the computation for arbitrary product initial states.

# 3 Entanglement dynamics of product states

In this section we will first provide a tensor network representation of the reduced density matrix for initial product states which allows both for an effective numerical computation even for large system size and for an analytic evaluation. Using this description we will then compute the Rényi entropies for different classes of impurity interactions.

## 3.1 Tensor network representation of the reduced density matrix for product states

We shall introduce a tensor network representations of the initial and time evolved states in this section first, before moving on to the discussion of the reduced density matrix.

### 3.1.1 Initial state

Let us begin by choosing two normalized states denoted by $|a\rangle$ and $|\circ\rangle$. Without loss of generality, we might choose $|\circ\rangle = |0\rangle$ as one of the computational basis states. Unless stated otherwise we take $|a\rangle$ orthogonal to $|\circ\rangle$. Diagrammatically, these two states are represented as, $|a\rangle = \!\!\begin{smallmatrix}\bullet\\a\end{smallmatrix}$ and $|\circ\rangle = \circ$.

We consider initial product states which are homogeneous in the bulk and correspond to $|a\rangle$ at the boundary. More precisely they are given by

$$|a_0\rangle = |a\rangle \otimes |\circ\rangle^{\otimes L} = \underset{a}{\bullet} \quad \circ \quad \circ \quad \cdots \quad \circ \quad \circ \quad \circ \in (\mathbb{C}^q)^{\otimes L+1} \, . \tag{17}$$

### 3.1.2 Time evolved state

Following the construction introduced in Ref. [65] to integrate out the free bulk dynamics of the boundary chaos circuit, we can recast the time evolved state into a more convenient tensor network of smaller size (by a factor $1/L$ compared to the naive tensor network representation). We provide a short description of the construction here. Let us start by introducing the building blocks of the network. We express time $t$ as $t = \tau L + \delta$ for non-negative integer $\tau$ and remainder $\delta \in \{0, 1, \dots, L-1\}$. For finite $L$, two different scenarios appear during evolution, the times $t$ with $l/2 < \delta$ and $l/2 < L - \delta$ are referred to as non-resonant and the remaining $t$ are called resonant. For resonant times $t/L$ differs from the closest integer by less than $l/(2L)$. The tensor network is composed from the local 2-qudit gates $V = UP$ defined by Eq. (8). We depict the gate $V$ and its Hermitian adjoint as

$$V = \blacksquare, \quad V^\dagger = \blacksquare, \tag{18}$$

where unitarity of $V$ diagrammatically reads

$$\blacksquare = \blacksquare = \Big| \Big| \tag{19}$$

The initial and final free dynamics for lattice sites in the bulk is taken into account by combining the action of the swap gates which for a given time $t$ are not connected to the boundary in forward or backward time direction into a global permutation of lattice sites. To this end we denote the unitary representation of the symmetric group on $L$ elements $S_L$ which permutes tensor factors by $\mathbb{P} : S_L \to \mathrm{U}\big((\mathbb{C}^q)^{\otimes L}\big)$. In other words, $\mathbb{P}_\sigma$ acts as $\mathbb{P}_\sigma\big(\bigotimes_{i=1}^L |\alpha_i\rangle\big) = \bigotimes_{i=1}^L |\alpha_{\sigma^{-1}(i)}\rangle$ on the canonical product basis. This is diagrammatically represented as,

$$\mathbb{P}_\sigma = \boxed{\phantom{xxxx}\mathbb{P}_\sigma\phantom{xxxx}} \tag{20}$$

Of particular importance for our construction are the permutations $\sigma_\delta \in S_L$ for $\delta \in \{0, 1, \ldots, L-1\}$ which are defined by their action on $x \in \{1, 2, \ldots, L\}$. To this end we write the latter as $x = (L - \delta + y) \mod L$ for unique $y \in \{1, 2, \ldots, L\}$ and define

$$\sigma_\delta(x) = \begin{cases} 2y - 1, & \text{if } 2y - 1 \le L, \\ 2(L - y + 1), & \text{if } 2y - 1 > L, \end{cases} \tag{21}$$

for $x \in \{1, 2, \ldots, L\}$. The tensor network representation of $|a_0(t)\rangle$ is given by (see Ref. [65] for details)

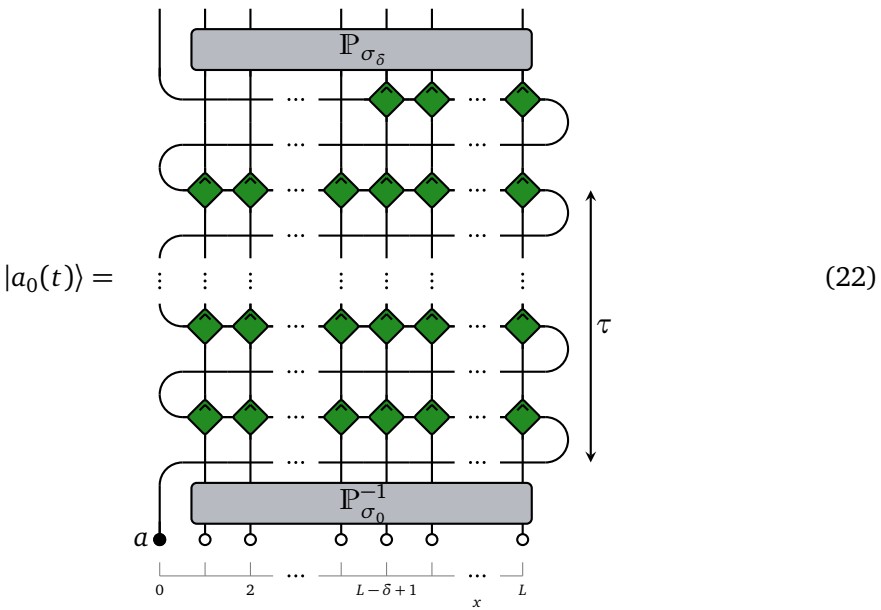

$$|a_0(t)\rangle = \qquad\qquad\qquad\qquad\qquad\qquad\qquad\qquad\qquad\qquad\qquad\qquad (22)$$

Note that the permutations (21) differ from Ref. [65] due to states evolving in the Schrödinger picture in contrast with operators evolving in the Heisenberg picture, which is manifest in the impurity interaction acting either in the second layer, see Eq. (7), or in the first layer of the circuit, see Eq. (12). For our choice of initial state $\left( \mathbb{1}_Q \otimes \mathbb{P}_{\sigma_0}^{-1} \right) |a_0\rangle = |a_0\rangle$ and we can replace $\mathbb{P}_{\sigma_0}^{-1}$ by the identity. A similar representation can be obtained for $\langle a_0(t)|$ in which the appropriately oriented adjoint gate $V^\dagger$ enters. Intuitively, in the above network evolution in the time-like variable $\tau$, i.e. vertically, describes scattering of excitations into the system with trivial free dynamics in between. Hence columns of the network describe such scattering events of this type from impurity interactions which differ by $L$ layers of the original circuit $\mathcal{U}^t$ obtained from Eq. (7). In a dual picture one might think of contracting the network in the horizontal spatial direction, which corresponds to scattering of excitations along the boundary. Consequently, rows of the tensor network (22) describe collective scattering events along the boundary from impurity interactions in $L$ subsequent layers in $\mathcal{U}^t$. Given the above interpretation, the physical time variable $t$ runs along a helix through the network and hence causes the helix-like topology of the tensor network.

### 3.1.3 Reduced density matrix

We now obtain the representation for the reduced density matrix from Eq. (22). We focus on the simpler case of non-resonant times. Expanding the reduced density matrix in the appropriate computational basis,

$$\rho_l(t) = \sum_{\alpha_0, \ldots, \alpha_l = 0}^{q-1} \sum_{\beta_0, \ldots, \beta_l = 0}^{q-1} \rho_{\beta_0 \cdots \beta_l}^{\alpha_0 \cdots \alpha_l}(t) |\alpha_0 \cdots \alpha_l\rangle \langle \beta_0 \cdots \beta_l|, \tag{23}$$

we can diagrammatically represent it as

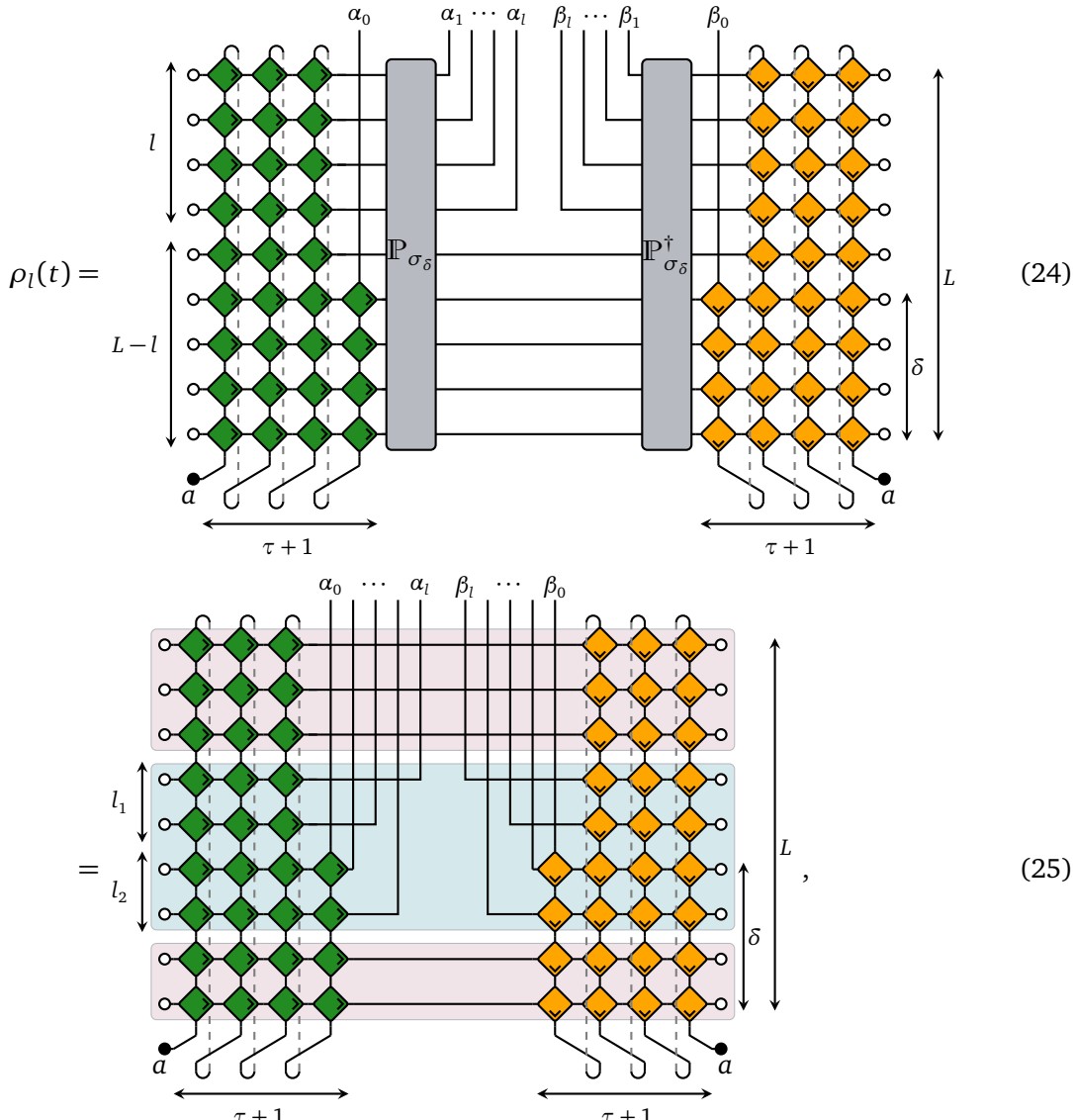

where $\alpha_0, \ldots, \alpha_l$ and $\beta_0, \ldots, \beta_l$ represent the output and input legs of $\rho_l(t)$ respectively. For a more convenient depiction of the diagrams we rotated the tensor network (22) by 90°. In order not to complicate the diagrams, we show it for fixed values of $L = 9$, $l = 4$ and $t = 31$. To obtain Eq. (24), we have used $\left(\mathbb{1}_Q \otimes \mathbb{P}_{\sigma_0}^{-1}\right)|a_0\rangle = |a_0\rangle$ and simplified the lowest row of the tensor network (22) (similarly for $\langle a_0(t)|$). Finally, Eq. (25) follows from the definition of $\sigma_\delta$ and the unitarity of $\mathbb{P}_{\sigma_\delta}$. Moreover, we define $l_1 = \lfloor l/2 \rfloor$ and $l_2 = l - l_1$. In the diagram, we also highlight the parts directly unconnected to the in- and output legs of the reduced density matrix with the rose shade, while we indicate the connected parts via the turquoise shade. The importance of this distinction will become clear in what follows.

To get an explicit expression of the reduced density matrix, we introduce different transfer matrices, which correspond to the rows of the tensor network (25). Hence, the transfer matrices act in the spatial direction corresponding to the vertical direction in Eq. (25) (and to the horizontal $x$-direction in Eq. (22)) Conceptually, this might be thought of as a dual description of the dynamics after a space-time swap, which was recently used in related contexts [16, 27, 30–32]. Formally, we define transfer matrices $\mathcal{T}_\tau$ and $[\mathcal{T}_\tau]_\beta^\alpha : (\mathbb{C}^q)^{\otimes 2\tau} \to (\mathbb{C}^q)^{\otimes 2\tau}$

for $\tau \geq 0$ as well as $[\mathcal{T}_\tau]^{\alpha_0 \alpha_1}_{\beta_0 \beta_1}$ and $[\mathcal{A}_\tau]^\alpha_\beta : (\mathbb{C}^q)^{\otimes 2\tau} \to (\mathbb{C}^q)^{\otimes 2(\tau-1)}$ for $\tau \geq 1$ as matrix product operators by their respective diagrammatic representation

$$\mathcal{T}_\tau = \text{ ○◆◆◆——◆◆◆○ },\tag{26}$$

$$[\mathcal{T}_\tau]^\alpha_\beta = \text{ ○◆◆◆•}\overset{\alpha\;\beta}{\phantom{x}}\text{•◆◆◆○ },\tag{27}$$

$$[\mathcal{T}_\tau]^{\alpha_0 \alpha_1}_{\beta_0 \beta_1} = \text{ ○◆◆◆•}\overset{\alpha_0 \qquad \beta_0}{\underset{\alpha_1 \beta_1}{\phantom{x}}}\text{•◆◆◆○ },\tag{28}$$

$$[\mathcal{A}_\tau]^\alpha_\beta = \text{ ○◆◆◆}\overset{\alpha \qquad \beta}{\phantom{x}}\text{◆◆◆○ }.\tag{29}$$

We also define $\mathcal{C}_{a,\tau} : (\mathbb{C}^q)^{\otimes 2\tau} \to (\mathbb{C}^q)^{\otimes 2(\tau+1)}, |\nu\rangle \mapsto |a\rangle \otimes |\nu\rangle \otimes |a\rangle$ which diagramatically can be expressed as

$$\mathcal{C}_{a,\tau} = \qquad\qquad\qquad\qquad\qquad\qquad\qquad\qquad\qquad\qquad\tag{30}$$

Additionally, we introduce $[\mathcal{A}_\tau]^{\alpha_0,\alpha_1}_{\beta_0,\beta_1} = [\mathcal{T}_\tau]^{\alpha_0,\alpha_1}_{\beta_0,\beta_1}$ for $l = 1$ and $[\mathcal{A}_\tau]^{\alpha_0\cdots\alpha_l}_{\beta_0\cdots\beta_l} : (\mathbb{C}^q)^{\otimes 2\tau} \to (\mathbb{C}^q)^{\otimes 2(\tau-1)}$ for $l \geq 2$ by

$$[\mathcal{A}_\tau]^{\alpha_0\cdots\alpha_l}_{\beta_0\cdots\beta_l} = \begin{cases} [\mathcal{T}_{\tau-1}]^{\alpha_l}_{\beta_l} \cdots [\mathcal{T}_{\tau-1}]^{\alpha_4}_{\beta_4} [\mathcal{T}_{\tau-1}]^{\alpha_2}_{\beta_2} [\mathcal{T}_\tau]^{\alpha_0 \alpha_1}_{\beta_0 \beta_1} \cdots [\mathcal{T}_\tau]^{\alpha_{l-1}}_{\beta_{l-1}}, & l \text{ even}, \\ [\mathcal{T}_{\tau-1}]^{\alpha_{l-1}}_{\beta_{l-1}} \cdots [\mathcal{T}_{\tau-1}]^{\alpha_4}_{\beta_4} [\mathcal{T}_{\tau-1}]^{\alpha_2}_{\beta_2} [\mathcal{T}_\tau]^{\alpha_0 \alpha_1}_{\beta_0 \beta_1} \cdots [\mathcal{T}_\tau]^{\alpha_l}_{\beta_l}, & l \text{ odd}. \end{cases}\tag{31}$$

The operators $\mathcal{A}_{\tau+1}$ represent the turquoise shaded part of the tensor network (25); while the lower rose shaded part corresponds to $[\mathcal{T}_{\tau+1}]^{\delta-l_2}$ and the upper rose shaded part corresponds to $\mathcal{T}_\tau^{L-\delta-l_1}$. With the above definitions we have,

$$\rho^{\alpha_0\cdots\alpha_l}_{\beta_0\cdots\beta_l}(t) = \text{tr}\left( \mathcal{T}_\tau^{L-\delta-l_1} [\mathcal{A}_{\tau+1}]^{\alpha_0\cdots\alpha_l}_{\beta_0\cdots\beta_l} \mathcal{T}_{\tau+1}^{\delta-l_2} \mathcal{C}_{a,\tau} \right).\tag{32}$$

We focus on the limit $L-\delta, \delta \gg l$ where Eq. (32) can be simplified further, as then the leading eigenvalues of $\mathcal{T}_{\tau+1}$ and $\mathcal{T}_\tau$ will give the dominant contribution. More precisely, we compute $\lim_{L,t\to\infty} \rho_l(t)$ for fixed $l$ while $L$ and $t$ approaching infinity such that $t/L \to \tau_0 \in \mathbb{R} \setminus \mathbb{Z}$. This latter condition ensures that for sufficiently large $L$ and $t$ we need to consider the non-resonant case only. In the above limit the resonant case is relevant only if $\tau_0 \in \mathbb{Z}$.

Using the unitarity of gates $V$ we can already list some basic properties of the spectrum of the transfer matrices. First note, that the transfer matrices are in general not normal, implying a nontrivial Jordan structure and a distinction between left and right eigenvectors. Nevertheless, the transfer matrices $\mathcal{T}_\tau$ are non-expanding [62] such that the leading eigenvalue is at most of modulus 1.

The leading eigenvalue of the transfer matrix is in fact equal to 1, which can be seen as follows. We first define the normalized rainbow states $|r_\tau\rangle \in (\mathbb{C}^q)^{\otimes 2\tau}$ via

$$|r_\tau\rangle = q^{-\frac{\tau}{2}} \sum_{\alpha_1,\ldots,\alpha_\tau=0}^{q-1} |\alpha_1 \alpha_2 \cdots \alpha_\tau \alpha_\tau \cdots \alpha_2 \alpha_1\rangle$$
$$= q^{-\frac{\tau}{2}} \qquad\qquad\qquad\qquad\qquad\qquad\qquad\tag{33}$$

By unitarity of the gates one has $\langle r_\tau | \mathcal{T}_\tau = \langle r_\tau |$ and hence $\langle r_\tau | = |r_\tau\rangle^\dagger$ is a left eigenvector for eigenvalue 1, as can be seen by evaluating the eigenvalue equation diagrammatically. The corresponding right eigenvector, however, cannot be described explicitly. In general, there might be more unimodular eigenvalues. However, all the unimodular eigenvalues, and in particular the eigenvalue 1, have equal algebraic and geometric multiplicity. The latter follows from observing that a non-trivial Jordan block corresponding to an unimodular eigenvalue of $\mathcal{T}_\tau$ is no longer non-expanding.

In any case, the above tensor network representation allows us to numerically study very large systems. The computational complexity to compute the reduced density matrix is linear in $L$ but exponential in $\tau$ and $l$ as the dimensions of the involved matrices go up to $q^{2(\tau+l+1)}$. However, additional constraints on the impurity interaction can lead to situations in which the reduced density matrix or the corresponding entanglement entropies can be computed analytically. In the following sections we shall use these ideas to compute entanglement growth in the boundary chaos circuit both analytically in the limit of large system size and long times $L, t \to \infty$ at fixed $\tau$. We complement those results by numerical computations in large but finite systems.

## 3.2 Entanglement dynamics

In this section, we use Eq. (32) to compute the growth of entanglement for different classes of impurities.

### 3.2.1 Impurities with a vacuum state

We start our analysis with a class of impurity interactions which allow for an exact computation of the reduced density matrix in the non-resonant case as $t, L \to \infty$. More precisely, we consider impurities which preserve a 2-qudit product state. We call this product state a (local) vacuum state. A trivial physical realization of such impurities, resulting in single-particle dynamics, is given by 2-qubit gates which exhibit a local U(1) symmetry, for which, e.g., magnetization is conserved. Hence either of the states $|00\rangle$ and $|11\rangle$ gives rise to a local vacuum state. However, in order to obtain a non-trivial dynamics, we consider generic vacuum-preserving gates described below.

Consider a two qudit gate $U \in \mathrm{U}(q^2)$ which has an eigenstate of product form $|\phi\rangle \otimes |\phi\rangle$, i.e.,

$$U |\phi\rangle \otimes |\phi\rangle = \mathrm{e}^{\mathrm{i}\varphi} |\phi\rangle \otimes |\phi\rangle \,. \tag{34}$$

Hence, $|\phi\rangle \otimes |\phi\rangle$ can be taken as the local vacuum state. The resulting circuit is equivalent (via local unitaries) to a circuit built from $\mathrm{e}^{\mathrm{i}\varphi} U_0$, where $U_0$ is block diagonal, i.e., $U_0 = 1 \oplus u$ with $u \in \mathrm{U}(q^2-1)$. As forward and backward time evolution appear symmetrically in the reduced density matrix, we can assume $\varphi = 0$ without loss of generality. We find such a system to be quantum chaotic in the spectral sense as numerically we find the circuit $\mathcal{U}$ built from such gates to exhibit level repulsion for generic choices of $u$, see Fig. 6 in App. A. Finally to simplify notation, after a potential change of the local basis we write $|\phi\rangle = |0\rangle = |\circ\rangle$ and denote the vacuum state by $|\circ\circ\rangle$.

**Spectrum of transfer matrices:** We shall now try to obtain the leading part of the spectrum of $\mathcal{T}$ built from gate $U = U_0$ placed at the left end of the circuit. As mentioned before, we intend to focus on the limit where the leading eigenvalues of $\mathcal{T}$ will give the dominant contribution to Eq. (32). We restrict ourselves to gates $U$ such that there are no additional unimodular eigenvalues of $\mathcal{T}_\tau$, except for eigenvalue 1 with multiplicity one. We call such gates *completely*

*chaotic* [62]. Numerically, we find this to be the case for generic choices of $u \in U(q^2 - 1)$, see App. B.

To compute the eigenvector corresponding to the leading eigenvalue 1, we first observe that $\mathcal{U} |\circ\rangle^{\otimes L+1} = |\circ\rangle^{\otimes L+1}$ as a consequence of $U |\circ\circ\rangle = |\circ\circ\rangle$ and $P |\circ\circ\rangle = |\circ\circ\rangle$. We also have $V |\circ\circ\rangle = |\circ\circ\rangle$, (and similar for $V^\dagger$) which can be diagrammatically expressed as

$$\text{(diagram)} = \text{(diagram)}, \quad \text{(diagram)} = \text{(diagram)}, \quad \text{(diagram)} = \text{(diagram)}, \quad \text{(diagram)} = \text{(diagram)}. \tag{35}$$

These imply that $\mathcal{T}_\tau |\circ\rangle^{\otimes 2\tau} = |\circ\rangle^{\otimes 2\tau}$, i.e., $|\circ\rangle^{\otimes 2\tau}$ is a right eigenvector for eigenvalue 1. The projection onto the eigenspace for eigenvalue 1 of $\mathcal{T}_\tau$ consequently reads

$$\mathcal{P}_\tau = q^{\frac{\tau}{2}} \left( |\circ\rangle^{\otimes 2\tau} \right) \langle r_\tau | , \tag{36}$$

where the prefactor takes into account the orthonormality with the left eigenvector $\langle r_\tau |$ defined in Eq. (33), required for the projector property $\mathcal{P}_\tau^2 = \mathcal{P}_\tau$. We shall compute the asymptotic reduced density matrix using Eq. (36).

**Asymptotic reduced density matrix:**   In the limit of large $L$ and hence $L - \delta \gg l_1$, we replace $\mathcal{T}_\tau^{L-\delta-l_1}$ by $\mathcal{P}_\tau$ in Eq. (32) and obtain,

$$\rho_{\beta_0 \cdots \beta_l}^{\alpha_0 \cdots \alpha_l}(t) = q^{\frac{\tau}{2}} \langle r_\tau | (\mathcal{A}_{\tau+1})_{\beta_0 \cdots \beta_l}^{\alpha_0 \cdots \alpha_l} (\mathcal{T}_{\tau+1})^{\delta - l_2} \left( |a\rangle \otimes |\circ\rangle^{\otimes 2\tau} \otimes |a\rangle \right) , \tag{37}$$

where[2] we have used the explicit definition of $\mathcal{C}_{a,\tau}$. Next, we consider also $\delta \gg l_2$ and replace $\mathcal{T}_{\tau+1}^{\delta - l_2}$ by $\mathcal{P}_{\tau+1}$ to obtain,

$$\rho_{\beta_0 \cdots \beta_l}^{\alpha_0 \cdots \alpha_l}(t) = q^{\frac{\tau}{2}} \langle r_\tau | (\mathcal{A}_{\tau+1})_{\beta_0 \cdots \beta_l}^{\alpha_0 \cdots \alpha_l} |\circ\rangle^{\otimes 2\tau+2} , \tag{38}$$

where we have used $q^{\frac{\tau+1}{2}} \langle r_{\tau+1} | \left( |a\rangle \otimes |\circ\rangle^{\otimes 2\tau} \otimes |a\rangle \right) = 1$. The invariance of the vacuum state implies

$$(\mathcal{A}_{\tau+1})_{\beta_0 \cdots \beta_l}^{\alpha_0 \cdots \alpha_l} |\circ\rangle^{\otimes 2\tau+2} = \left( \prod_{i=0}^{l} \delta_{\alpha_i,0} \delta_{\beta_i,0} \right) |\circ\rangle^{\otimes 2\tau} . \tag{39}$$

Combining this with $q^{\frac{\tau}{2}} \langle r_\tau | \left( |\circ\rangle^{\otimes 2\tau} \right) = 1$ we obtain $\rho_{\beta_0 \cdots \beta_l}^{\alpha_0 \cdots \alpha_l}(t) = \prod_{i=0}^{l} \delta_{\alpha_i,0} \delta_{\beta_i,0}$. Hence, we get,[3]

$$\rho_l(t) = (|\circ\rangle\langle\circ|)^{\otimes l+1} + c(\tau, l) \lambda_0^\delta . \tag{40}$$

Here we explicitly include subleading terms which scale with the subleading eigenvalue $\lambda_0 = \lambda_0(\tau)$ of $\mathcal{T}_{\tau+1}$, which in general depends on $\tau$. The prefactor can in principle be obtained from the left and right eigenvectors corresponding to $\lambda_0$. Further note, that due to biorthogonality of left and right eigenvectors and Eq. (39) the subleading part of the spectrum of $\mathcal{T}_\tau$ gives rise to contributions exponentially suppressed with $L$ only, which can safely be ignored in Eq. (40).

---

[2]Strictly speaking the above expression is correct only in the limit $L \to \infty$ or up to corrections exponentially suppressed with $L - \delta$ or $\delta$. For the sake of notational convenience we still write it as an equality. We discuss subleading terms explicitly when appropriate.

[3]Alternatively, we could have replaced $\mathcal{T}_{\tau+1}$ first, but the subleading contribution would be still be the same. This is because, replacing $\mathcal{T}_{\tau+1}$ by $\mathcal{P}_{\tau+1}$ and subsequently $\mathcal{T}_\tau$ by the projection onto $\lambda_0$ eigenspace gives zero by Eq. (39) and biorthogonality of eigenvectors.

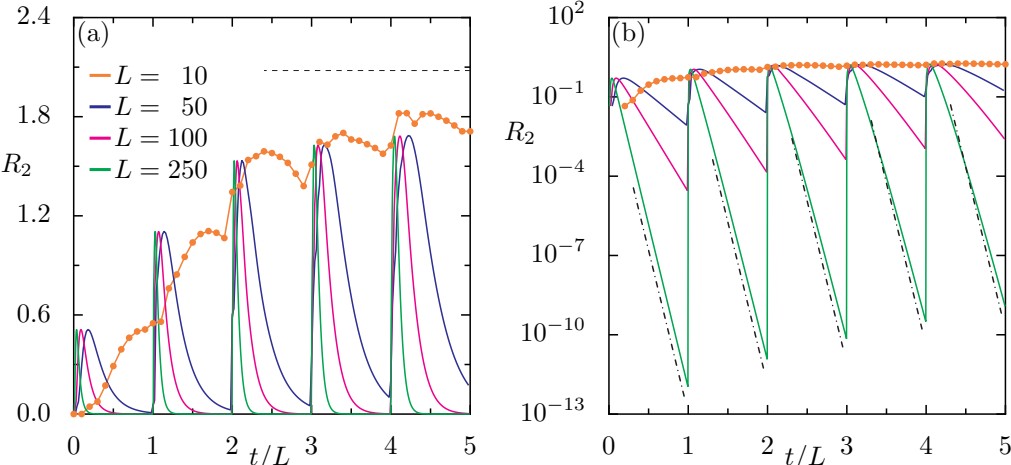

Figure 1: Second Rényi entropy for $q = 2$ and $l = 2$ for an impurity interaction with a vacuum state for various system sizes in (a) linear and (b) semi-logarithmic scale. (a) The dashed line corresponds to the maximum entropy given by $(l+1)\ln(q)$. Orange dots depict $R_2$ obtained from a direct computation via Eq. (14). (b) The dash-dotted lines illustrates the asymptotic scaling $|\lambda_0|^\delta$.

**Entanglement entropies and comparison with numerics:** From Eq. (40), the Rényi entropies follow as

$$R_n(t) \sim \frac{n}{n-1}|\lambda_0|^\delta \,, \tag{41}$$

assuming unique subleading eigenvalue and ignoring possible non-trivial Jordan blocks, as both do not change the result qualitatively. The above implies entanglement entropies to be exponentially suppressed with $\delta$. Hence entanglement can be large only when $\delta$ is small implying persistent revivals of entanglement entropies with period given by the system size $L$. This is illustrated in Fig. 1 for the second Rényi entropy. There we show the entanglement entropies obtained from numerically evaluating Eq. (32) for various large system sizes. In particular, we confirm the asysmptotic scaling $|\lambda_0|^\delta$ in Fig. 1(b). We do not depict Rényi entropies of higher order $n > 2$ as they are practically indistinguishable from $n = 2$. For small system sizes, for which entanglement dynamics can be evaluated directly, i.e., by performing time evolution with the original circuit $\mathcal{U}$, Eq. (7), we find saturation of the entanglement entropies at late times (not shown). The saturation value, however, is in general not that of a random state given by the Page value [70,71], but depends on the concrete choice of the gate.

### 3.2.2 T-dual impurities

Another situation in which the leading part of the spectrum can be described explicitly is given by T-dual impurity interactions at the boundary. In what follows we shall elaborate on the implications of T-duality of the impurity interaction on the entanglement of states.

**Spectrum of transfer matrices:** A 2-qudit gate $U \in \mathrm{U}(q^2)$ is called T-dual if its partial transpose $U^{\mathrm{T}_1}$ with respect to the first qudit (and hence also w.r.t. the second qudit) is unitary as well [72]. A convenient parameterization of T-dual gates is given by [73,74]

$$U = (u_+ \otimes u_-)\exp\big(\mathrm{i}J\Sigma_{q^2-1} \otimes \Sigma_{q^2-1}\big)(v_+ \otimes v_-) \,. \tag{42}$$

with $\Sigma_i$ the generalized Gell-Mann matrices, $J \in [0, \pi/4]$ and $u_\pm, v_\pm \in U(q)$. Here, $J$ governs the interaction with $J = \pi/4$ giving rise to the most rapidly decaying correlations and chaotic dynamics. The above is an exhaustive parametrization for $q = 2$ but not for $q > 2$ [73]. Consequently the gate $V$ becomes dual unitary, i.e., the gate $\tilde{V}$ which originates from $V$ by reshuffling of matrix elements (w.r.t the canonical product basis) according to $\tilde{V}^{ab}_{cd} = V^{db}_{ca}$ is unitary [73]. This can be diagrammatically expressed as

$$
\begin{array}{c} \text{(diagram)} \end{array} \tag{43}
$$

Denoting by $|\circ\rangle \in \mathbb{C}^q$ an arbitrary normalized state as boundary condition for the transfer matrices $\mathcal{T}_\tau$ dual unitarity implies that the rainbow state $|r_\tau\rangle$ is a right eigenvector of $\mathcal{T}_\tau$ with eigenvalue 1, i.e., $\mathcal{T}_\tau |r_\tau\rangle = |r_\tau\rangle$. Hence the left and right eigenvectors coincide in this case. In what follows, we consider only those T-dual impurity interactions where eigenvalue 1 has multiplicity 1 and no other unimodular eigenvalues exist, [4] i.e., the completely chaotic gates. Note, that the set of T-dual gates and the set of gates which support a local vacuum are not disjoint. However, for qubits $q = 2$, the gate implementing impurity interaction which share both features has to be of the form $U = |\circ\rangle\langle\circ| \otimes v + |a\rangle\langle a| \otimes w$ where $v$ is a diagonal (phase) unitary single qubit gate, or a similar expression with the first and second qubit swapped. Such interactions cannot create entanglement for the specified initial states. The form above follows directly from demanding unitarity of $U^{T_1}$ for gates of the form $U = 1 \oplus u$ with $u \in U(3)$. In contrast, for larger local Hilbert space dimensions, $q > 2$, there exist gates which are both T-dual and exhibit a vacuum state, which give rise to non-trivial entanglement dynamics. Examples for those gates are the folded gates described in Sec. 2.1 when the unfolded gate is T-dual. This leads to the entanglement dynamics described in Sec. 4.2.2 for local operators.

**Asymptotic Reduced Density Matrix:** The above properties allow two construct the asymptotic form of the reduced density matrix in the non-resonant case ($L, t \to \infty$, $t/L \to \tau_0 \in \mathbb{R} \setminus \mathbb{Z}$) for initial states of the form Eq. (17). That is we start from $|a\rangle \otimes |\circ\rangle^{\otimes L}$ with $|a\rangle \in \mathbb{C}^q$ an arbitrary normalized state (not necessarily orthogonal to $|\circ\rangle$). Note, that as $|r_\tau\rangle$ is a right eigenvector of $\mathcal{T}_\tau$ independently of the choice of boundary conditions, the following construction can be applied to arbitrary initial product states, when taking proper care of the action of $\mathbb{1}_q \otimes \mathbb{P}^{-1}_{\sigma_0}$ on the initial state. To simplify the discussion, however, we restrict to the simpler initial states above. The asymptotic reduced density matrix is obtained by replacing powers of $\mathcal{T}_\tau$ with $|r_\tau\rangle\langle r_\tau|$ and powers of $\mathcal{T}_{\tau+1}$ with $|r_{\tau+1}\rangle\langle r_{\tau+1}|$. This yields

$$
\rho^{\alpha_0 \cdots \alpha_l}_{\beta_0 \cdots \beta_l}(t) = \langle r_{\tau+1} | (|a\rangle \otimes |r_\tau\rangle \otimes |a\rangle) \; \langle r_\tau | (\mathcal{A}_{\tau+1})^{\alpha_0 \cdots \alpha_l}_{\beta_0 \cdots \beta_l} |r_{\tau+1}\rangle \tag{44}
$$

$$
= q^{-\frac{1}{2}} \langle r_\tau | (\mathcal{A}_{\tau+1})^{\alpha_0 \cdots \alpha_l}_{\beta_0 \cdots \beta_l} |r_{\tau+1}\rangle \,, \tag{45}
$$

---

[4] We have confirmed numerically that generically this is the case and that there are no other unimodular eigenvalues for all $\tau$ (and that there is a finite spectral gap $1 - |\lambda_0| > 0$ as $\tau \to \infty$), see. App. B.

up to subleading terms. In the last equality we have used $\langle r_{\tau+1}|(|a\rangle \otimes |r_\tau\rangle \otimes |a\rangle) = q^{-\frac{1}{2}}$. A diagrammatic representation of the asymptotic reduced density matrix is given by

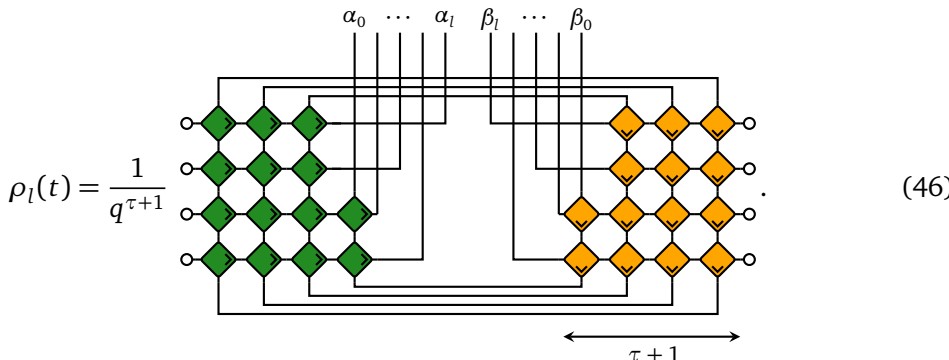

$$\rho_l(t) = \frac{1}{q^{\tau+1}} \qquad\qquad\qquad\qquad\qquad . \tag{46}$$

Unfortunately, unitarity and dual unitarity of $V$ does not allow to simplify the reduced density matrix further except for $l = 0$. However the Rényi entropies can still be computed in the asymptotic regime of large subsystem size, which is discussed below. Similar to the setting of gates with a vacuum state, subleading terms are at least suppressed as $\lambda_0^\delta$. However, numerical investigations, as presented in Fig. 2, seem to indicate that subleading terms are suppressed with system size, i.e., as $\lambda_0^L$.

**Rényi entropies and comparison with numerics:** The asymptotics of the Rényi entropies can be obtained when $L - \delta$, $\delta$ and $l$ are large. Formally, we consider first the simultaneous limit $L, t \to \infty$, $t/L \to \tau_0 \in \mathbb{R} \setminus \mathbb{Z}$ as described in Sec. 3.1.3, which gives the reduced density matrix derived in Sec. 3.2.2 and afterwards the limit $l \to \infty$.

The goal, is to write down the Rényi entropies in terms of the leading eigenvalues of the transfer matrices. To do so, we express $\mathrm{tr}(\rho_l(t)^n)$ in a form, in which the asymptotic approximation can be easily applied. More precisely, we aim to obtain $\mathrm{tr}(\rho_l(t)^n) \propto \langle \sigma_\tau | (\mathcal{T}_\tau^l)^{\otimes n} | \sigma_\tau \rangle$ for a suitable state $|\sigma_\tau\rangle \in (\mathbb{C}^{2\tau})^{\otimes n}$.

To this end, we first define

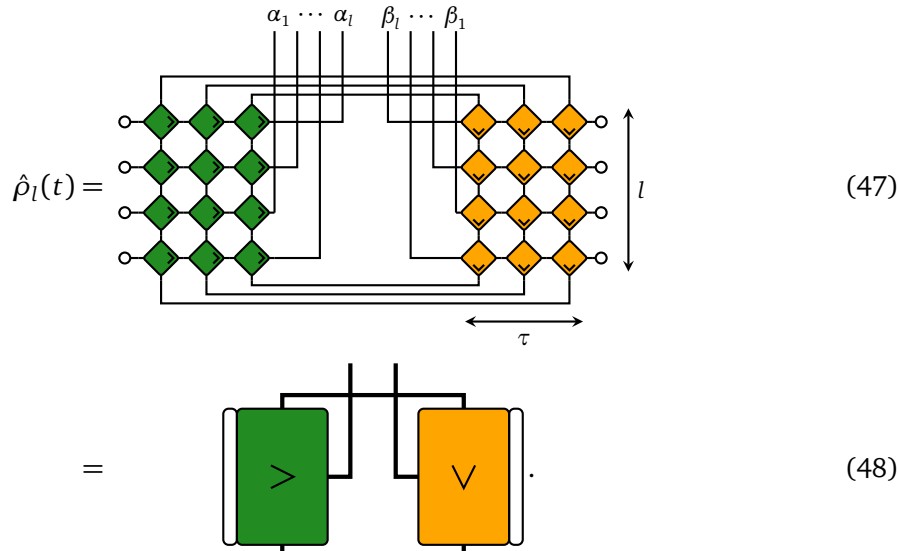

$$\hat{\rho}_l(t) = \qquad\qquad\qquad\qquad\qquad\qquad \tag{47}$$

$$= \qquad\qquad\qquad\qquad\qquad . \tag{48}$$

Note that, in Eq. (47) the number of in- and output legs is reduced by one. That is, $\hat{\rho}_l(t)$ is of dimension $q^l$ instead of $q^{l+1}$. Moreover, only transfer matrices $[\mathcal{T}_\tau]_\beta^\alpha$ enter, in contrast with Eq. (46) in which also transfer matrices at size $\tau + 1$ enter. The second line, Eq. (48), is a schematic representation used to make the diagrams more compact. The green and orange

colored blocks correspond to the $\tau \times l$ block built from the gates $V$ and $V^\dagger$, respectively, with the wedges indicating the orientation of the gates, while the white boxes indicate the boundary conditions given by $|\circ\rangle$. The wires corresponding to in and out going legs each carry a Hilbert space of dimension $q^l$. The top and bottom wires are an abbreviation for the unnormalized rainbow states $q^{\frac{\tau}{2}} |r_\tau\rangle$ and hence those wires carry a Hilbert space of dimension $q^\tau$. The above definitions allow us to rewrite

$$\mathrm{tr}(\rho_l(t)^n) = q q^{-n(\tau+1)} \mathrm{tr}(\hat{\rho}_l(t)^n)\,, \tag{49}$$

where a factor $q^{-n(\tau+1)}$ enters due to $n$ normalization constants in Eq. (46) coming from $n$ replicas of $\rho_l(t)$. The first factor of $q$, however, originates from repeatedly contracting the gates $V$ connected to output legs $\alpha_0$ and $\alpha_1$ of the $i$-th replica with the adjoint gates $V^\dagger$ connected to the input legs $\beta_0$ and $\beta_1$ in the $i-1$-th replica using unitality of the gates. This removes the dependence of our results from $\delta$, such that entanglement entropies will depend only on $\tau$.

For $\tau = 0$, Eq. (49) can be evaluated exactly as $\hat{\rho}_l(t) = (|\circ\rangle\langle\circ|)^{\otimes l}$ even for finite $l$. This gives the initial entropy as

$$R_n(t) = \ln(q)\,, \tag{50}$$

up to terms exponentially suppressed in $\delta$, i.e. $\sim |\lambda_0|^\delta$, with $\lambda_0$ being the subleading eigenvalue of $\mathcal{T}_1$. This gives rise to non-trivial initial dynamics of the entanglement entropies for short times for any finite $l$, see Fig. 2, as well as in the limit $l \to \infty$.

For $\tau > 0$ the $n$ replicas entering $\mathrm{tr}(\rho_l(t)^n)$ in Eq. (49) need to be rearranged to proceed further. This is best seen schematically. For $n = 3$ (with an obvious generalization to arbitrary $n$) this reads

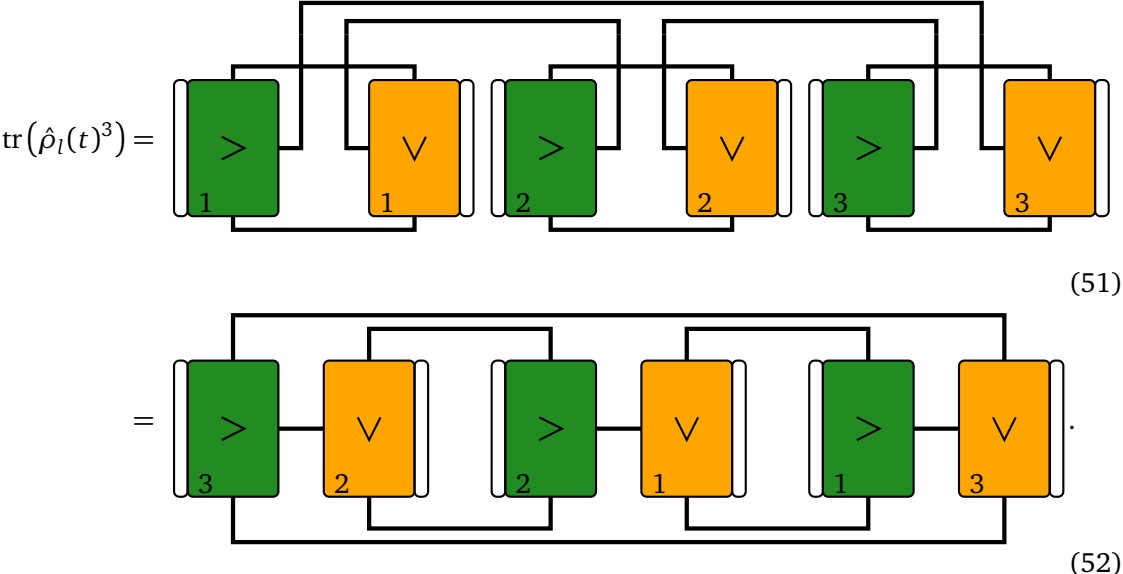

$$\tag{51}$$

$$\tag{52}$$

The first equality diagrammatically represents the multiplication of subsequent replicas by connecting the input legs of replica $i$ with the output legs of replica $i+1$ (see labels in the bottom left of the boxes) and connecting the legs between $n$ and $1$ realizes the trace. The second equality is obtained by rearranging the boxes corresponding to the forward and backward time evolution part while keeping the lines connecting subsequent boxes intact. Then, each combined block consisting of forward block (green) of replica $i$ and backward block (orange)

from replica $i-1$ is $\mathcal{T}_\tau^l$ i.e.,

$$\mathcal{T}_\tau^l = \quad \boxed{>}\!-\!\boxed{\vee} \tag{53}$$

The wires connecting the combined blocks on the top and bottom of the network can be viewed as states $|\boldsymbol{\sigma}^\tau\rangle \in (\mathbb{C}^q)^{\otimes 2n\tau}$. To give a proper definition we denote the $2n\tau$-periodic shift by $-\tau$ in $S_{2n\tau}$ by $\eta_{-\tau}$, and similar to Sec. 3, the unitary representation of $S_{2n\tau}$ which permutes the tensor factors in $(\mathbb{C}^q)^{\otimes 2n\tau}$ by $\mathbb{P}$. The states $|\boldsymbol{\sigma}^\tau\rangle$ are then defined by

$$|\boldsymbol{\sigma}^\tau\rangle = q^{\frac{n\tau}{2}} \mathbb{P}_{\eta_{-\tau}} |r_\tau\rangle^{\otimes n} \tag{54}$$

$$= \quad \underline{|\quad \underline{\lfloor\_\rfloor}\quad|\quad \underline{\lfloor\_\rfloor}\quad|\quad \underline{\lfloor\_\rfloor}\quad \cdots \quad \underline{\lfloor\_\rfloor}\quad|} \tag{55}$$

where each wire carries the Hilbert space $(\mathbb{C}^q)^{\otimes\tau}$ of dimension $q^\tau$. Evidently, $|\boldsymbol{\sigma}^\tau\rangle$ is just a shifted version of the $n$-fold tensor product of the unnormalized rainbow states, where we shift by "half a replica". Hence Eq. (52) can be phrased as

$$\text{tr}(\hat{\rho}_l(t)^n) = \langle \boldsymbol{\sigma}^\tau | \left(\mathcal{T}_\tau^l\right)^{\otimes n} |\boldsymbol{\sigma}^\tau\rangle . \tag{56}$$

The above expression can be simplified in the limit $l \to \infty$ by replacing $\mathcal{T}_\tau^l$ by the projection onto the leading eigenvalue, which by the assumption of having multiplicity 1 is given by $\mathcal{P}_\tau = |r_\tau\rangle\langle r_\tau|$. Using the diagrammtic representations of states, we find

$$\text{tr}(\hat{\rho}_l(t)^n) = \langle \boldsymbol{\sigma}^\tau | (|r_\tau\rangle\langle r_\tau|)^{\otimes n} |\boldsymbol{\sigma}^\tau\rangle = q^{-\tau(n-2)} . \tag{57}$$

Inserting the above result into Eq. (49) finally yields

$$\text{tr}\left(\rho_l^n(t)\right) = q^{-(n-1)(2\tau+1)} , \tag{58}$$

where the subleading terms are exponentially suppressed at least as $|\lambda_0|^l$. Consequently the corresponding Rényi entropies read

$$R_n(t) = 2\tau \ln(q) + \ln(q) . \tag{59}$$

For the resonant case, i.e. when $L, t \to \infty$, such that, $t/L \to \tau_0 \in \mathbb{Z}$ first and subsequently $l \to \infty$, a similar computation yields,

$$\text{tr}(\rho_l(t)^n) = q^{-(n-1)2\tau} , \tag{60}$$

and hence,

$$R_n(t) = 2\tau \ln(q) . \tag{61}$$

Note that the computation for the resonant case involves essentially the same steps but with slightly different intermediate tensor networks.

Even though the limit of large subsystem size is not accessible by numerical simulation, the staircase structure of entanglement entropies suggested by Eq. (59) is clearly seen in Fig. 2(b) for small $l = 2$. The average slope, however, is different from $2\ln(q)$ as in Eq. (59). We restrict ourselves to the second Rényi entropy $n = 2$ in Fig. 2(b) as higher orders $n > 2$ give qualitatively similar results. Small differences appear, however, for the value of the plateaus at fixed $\tau > 0$, which weakly depends on $n$.

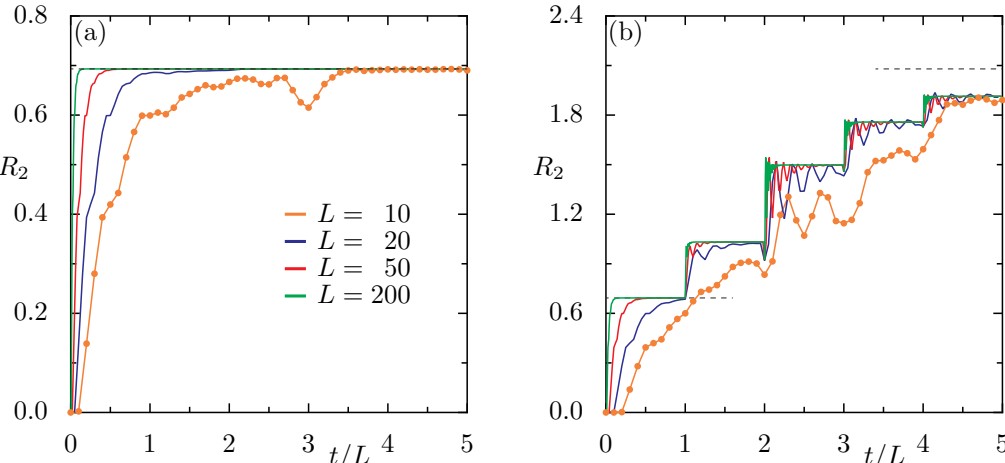

Figure 2: Second Rényi entropy for $q = 2$ with (a) $l = 0$ and (b) $l = 2$ for a T-dual impurity interaction for various system sizes. (a) The dashed line corresponds to the maximum entropy given by Eq. (63). Orange dots depict $R_2$ obtained from a direct computation via Eq. (14). We choose $J = \pi/4 - 0.05$ in Eq. (42) to ensure chaotic dynamics.

**Subsystem with one lattice site, $l = 0$:**   Another limit which can be treated exactly is that of the smallest possible subsystem given by $l = 0$, i.e., the subsystem $A$ consisting of the first lattice site only. We again consider the limit $t, L \to \infty$ and the simpler non-resonant case first. Applying the analysis to compute the reduced density matrix for T-dual gates to $l = 0$, we get the reduced density matrix as given by Eq. (45). Then, using $(\mathcal{A}_{\tau+1})^\alpha_\beta |r_{\tau+1}\rangle = q^{-\frac{1}{2}} \delta_{\alpha,\beta} |r_\tau\rangle$ we get,

$$(\rho_l(t))^\alpha_\beta = \frac{1}{q} \delta_{\alpha,\beta} = \frac{1}{q} \mathbb{1}_q \,, \tag{62}$$

which is the infinite temperature state. Consequently, the Rényi entropies read

$$R_n(t) = \ln(q) \,. \tag{63}$$

For the resonant case, a similar computation yields the same result. Moreover, as argued in the previous section, the above result is obtained for $\tau = 0$ as well, but with corrections scaling as $|\lambda_0|^\delta$. In particular, after the non-trivial initial dynamics, entanglement entropies saturate at the maximum possible value, as it is depicted in Fig. 2(a). For any other numerically accessible subsystem size this is not the case, see Fig. 2(b), even for longer times than what is shown there.

### 3.2.3   Numerical results for generic impurities

For impurity interactions falling in neither of the classes discussed above, there is no simple description of the right eigenvectors corresponding to leading eigenvalue 1. Nevertheless, the tensor network representation (25) allows for computing the reduced density matrix for large system size $L$ but small subsystem size $l$ numerically. Here we briefly report the numerical results. In Fig. 3 we depict the second Rényi entropy for (a) $l = 0$ and (b) $l = 2$. In both cases the entanglement dynamics resembles a combination of the T-dual case and the case of gates with local vacuum states. In particular we observe a similar staircase structure as in the T-dual case. This is induced by the leading eigenvalue 1 of the transfer matrices $\mathcal{T}_\tau$ and $\mathcal{T}_{\tau+1}$ and the corresponding eigenvectors. The latter give the reduced density matrix as $L \to \infty$ similar to the tensor network (46), with the rainbow state on the bottom of the network replaced by

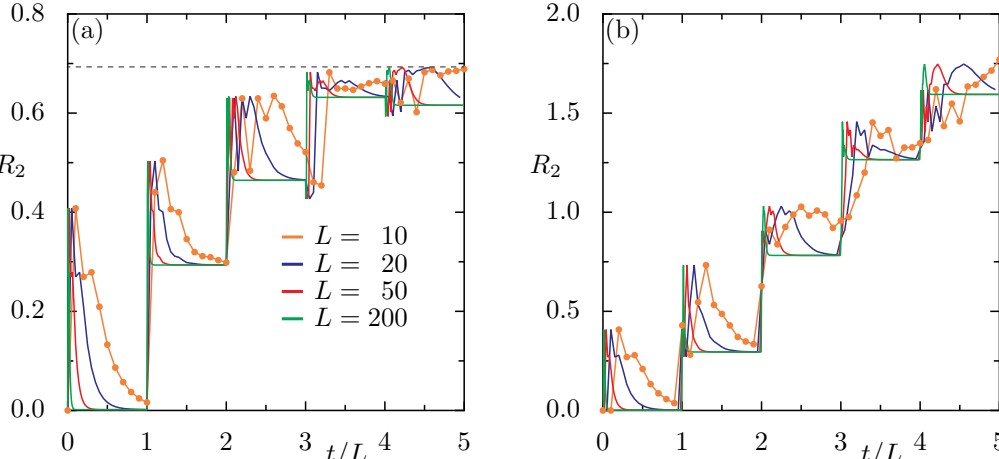

Figure 3: Second Rényi entropy for $q = 2$ with (a) $l = 0$ and (b) $l = 2$ for a generic impurity interaction for various system sizes. (a) The dashed line corresponds to the maximum entropy $(l + 1)\ln(q)$. Orange dots depict $R_2$ obtained from a direct computation via Eq. (14).

the actual (unknown) right eigenvector for the leading eigenvalue 1. The Rényi entropy of this asymptotic reduced density matrix gives rise to the plateaus observed for constant $\tau$. In principle this is also the case for impurity interactions which support a vacuum state. There, however, the asymptotic reduced density matrix further contracts to a pure state leading to a plateau of height zero. In contrast, perturbing gates with vacuum states one expects the right eigenvector for eigenvalue 1 to change as well as the vacuum states being no longer exactly invariant, both of which will lead to a plateau of non-zero height. On top of the plateaus we observe additional contributions which originates from the subleading eigenvalues and scale as $|\lambda_0|^\delta$. This is reminiscent of impurity interactions with vacuum states, for which those contributions sit on top of the plateaus of zero height as described above. For generic gates, the magnitude of these subleading contributions strongly depend on the concrete choice of impurity interaction. With non-zero probability we find both examples qualitatively similar to the one shown in Fig. 3 as well as examples for which subleading contributions are essentially irrelevant and the entanglement dynamics is very similar to the T-dual case. In any case we observe saturation of entanglement entropies for small system sizes at late times, i.e. longer than what is depicted here, but the maximum possible value of $(l + 1)\ln(q)$ is in general not reached. We also checked Rényi entropies of higher order $n > 2$ and find almost identical values, in particular there is a very weak dependence on $n$ of the height of the plateaus similar to the T-dual case.

# 4 Operator entanglement dynamics for local operators

In this section we study the entanglement dynamics of local initial operators. In analogy to the case of states we first construct a tensor network representation similar to Eq. (25) for the reduced super density matrix in Sec. 4.1. This again allows for an exact computation of the asymptotic reduced super density matrix in the limit $L, t \to \infty$ and subsequently of the operator Rényi entropies. We present this calculation for both generic impurity interactions in Sec. 4.2.1 and T-dual impurity interaction in Sec. 4.2.2.

## 4.1 Tensor network representation of the reduced super density matrix

We construct the analog of the tensor network representation (25) for the super density matrix for an initial local operator. With a slight abuse of notation we will use the same symbols and diagrammatic representations in the following sections as we did for the state counterparts in the previous sections, as many constructions and arguments are exactly the same for the operators as for the states. However, there are some notable differences, which are as follows.

Firstly, there are slight differences in the tensor networks representations which ultimately originate from the time evolution of states in the Schrödinger picture as opposed to the time evolution of operators in the Heisenberg picture. Those differences are essentially irrelevant for the dynamics of entanglement entropies. The major difference, however, is that the folded gates are unital (see definition below) which leads to additional properties of the relevant transfer matrices. In the following sections we will often drop the adjective 'super' when referring to super operators and super density matrices.

Let us first introduce the relevant notation and relate it to the constructions for states in the previous sections. This will provide us with the tensor network representation of the reduced density matrix. Subsequently, for different choices of impurity interactions we will compute its asymptotic form and derive the corresponding entanglement entropies.

The local Hilbert space is now the space of vectorized operators $\mathbb{C}^{q^2}$ of dimension $q^2$ with the Hilbert-Schmidt orthonormal basis $(|\alpha\rangle)_{\alpha=0}^{q^2-1}$ introduced in Sec. 2.1. We denote the Hilbert-Schmidt normalized vectorized identity $\mathbb{1}_q/\sqrt{q}$ by $|\circ\rangle = |0\rangle$ and choose a Hermitian and traceless Hilbert-Schmidt normalized vectorized operator $|a\rangle \in \mathbb{C}^{q^2}$ (being traceless implies $\langle\circ|a\rangle = 0$). We shall depict them diagrammatically in the same way as for states and hence, the corresponding local operator $|a_0\rangle = |a\rangle \otimes |\circ\rangle^{\otimes L} \in \left(\mathbb{C}^{q^2}\right)^{\otimes L+1}$ is diagrammatically presented exactly as in Eq. (17). We introduce the folded gate $W = U^\dagger \otimes U^T$ and $V = WS$ (which is the folded version of the gate $V = UP$ in the case of states) and use the same diagrammatic representation (18). The gate $V$ is again unitary, which is diagrammatically depicted by Eq. (19).

We define permutations $\sigma_\delta \in S_L$ for $\delta \in \{0, 1, \ldots, L-1\}$ similarly as in the case of states by its action on $x \in \{1, 2, \ldots, L\}$. Again we write the latter as $x = (L - \delta + y) \bmod L$ for unique $y \in \{1, 2, \ldots, L\}$ and define

$$\sigma_\delta(x) = \begin{cases} 2y, & \text{if } 2y \leq L, \\ 2(L-y)+1, & \text{if } 2y > L. \end{cases} \tag{64}$$

Redefining the permutations $\sigma_\delta$ is a consequence of the differences between the tensor network representations (7) and (12) reflecting evolution in the Schrödinger and the Heisenberg picture, respectively. Again, $\mathbb{P}$ is the unitary representation of $S_L$ permuting tensor factors, which now acts on $\left(\mathbb{C}^{q^2}\right)^{\otimes L}$. With the above notations, the time evolved local operator is given by Eq. (22), and is diagrammatically represented by the tensor network (24). Keeping in mind the difference in the permutations $\sigma_\delta$, we ultimately obtain a very similar tensor network representation as Eq. (25), given by

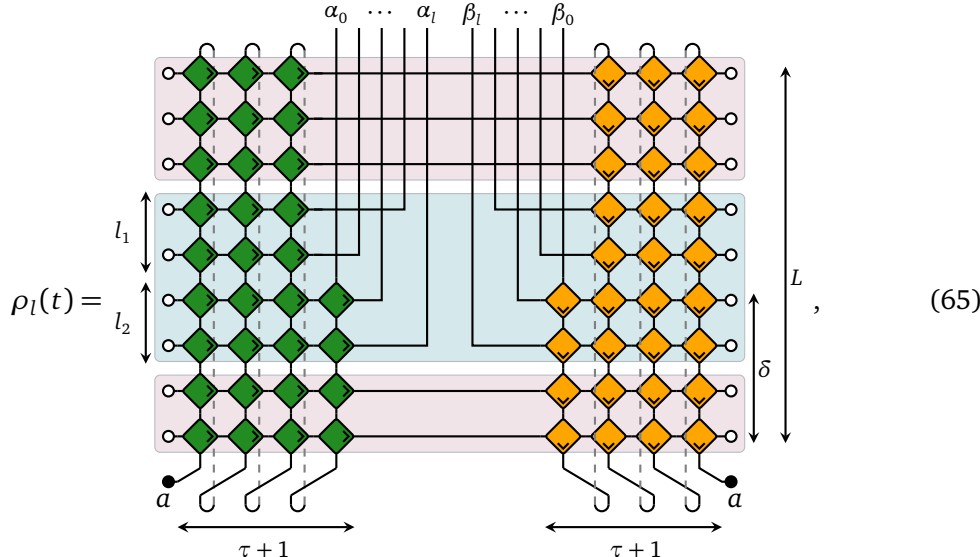

$$\rho_l(t) = \qquad\qquad , \qquad (65)$$

where $l_2 = \lfloor l/2 \rfloor$ and $l_1 = l - l_2$. The difference to Eq. (25) is just that even and odd in- and output legs of the reduced density matrices for sites 1 to $l$ are interchanged. The rows of the network can again be cast in the form of transfer matrices given by Eqs. (26)-(30), but we redefine $[\mathcal{A}_\tau]_{\beta_0\beta_1}^{\alpha_0\alpha_1} = [\mathcal{T}_{\tau-1}]_{\beta_1}^{\alpha_1} [\mathcal{A}_\tau]_{\beta_0}^{\alpha_0}$ for $l = 1$ as well as

$$[\mathcal{A}_\tau]_{\beta_0\cdots\beta_l}^{\alpha_0\cdots\alpha_l} = \begin{cases} [\mathcal{T}_{\tau-1}]_{\beta_{l-1}}^{\alpha_{l-1}} \cdots [\mathcal{T}_{\tau-1}]_{\beta_1}^{\alpha_1} [\mathcal{T}_\tau]_{\beta_0\beta_2}^{\alpha_0\alpha_2} [\mathcal{T}_\tau]_{\beta_4}^{\alpha_4} \cdots [\mathcal{T}_\tau]_{\beta_l}^{\alpha_l} \,, & l \text{ even}, \\ [\mathcal{T}_{\tau-1}]_{\beta_l}^{\alpha_l} \cdots [\mathcal{T}_{\tau-1}]_{\beta_1}^{\alpha_1} [\mathcal{T}_\tau]_{\beta_0\beta_2}^{\alpha_0\alpha_2} [\mathcal{T}_\tau]_{\beta_4}^{\alpha_4} \cdots [\mathcal{T}_\tau]_{\beta_{l-1}}^{\alpha_{l-1}} \,, & l \text{ odd}, \end{cases} \qquad (66)$$

for $l > 1$. From here on the same techniques can be applied to compute entanglement entropies as in the case of states.

## 4.2 Entanglement dynamics

Now we shall compute the entanglement dynamics for operators for different kinds of impurity interactions, and highlight the difference with the dynamics of states, if any.

### 4.2.1 Generic impurity interactions

In this section we study the entanglement dynamics for local operators in case of generic (completely chaotic, see below) unitary interactions. This is closely related to the case of gates with a vacuum state in Sec. 3.2.1, as the vectorized identity plays the role of the vacuum state.

In the case of operators the gate $V$ and its adjoint are unital, as they act by conjugation with the unitaries $UP$ or $PU^\dagger$ on operators, i.e.,

$$V \lvert \circ\circ \rangle = \lvert \circ\circ \rangle \qquad \text{and} \qquad V^\dagger \lvert \circ\circ \rangle = \lvert \circ\circ \rangle \,. \qquad (67)$$

By taking the adjoint of the above equations, one can see that unitality applies also to $\langle \circ\circ \rvert$, corresponding to trace preservation. Diagrammatically, this can be expressed as previously in Eq. (35). Consequently the transfer matrices are unital as well, i.e., $\mathcal{T}_\tau \lvert \circ \rangle^{\otimes 2\tau} = \lvert \circ \rangle^{\otimes 2\tau}$, meaning that $\lvert \circ \rangle^{\otimes 2\tau}$ is a right eigenvector for eigenvalue one. The above implies that $\mathcal{T}_\tau$ is a unital CP map and that $\mathcal{T}_\tau^\dagger$ is a CPTP map.

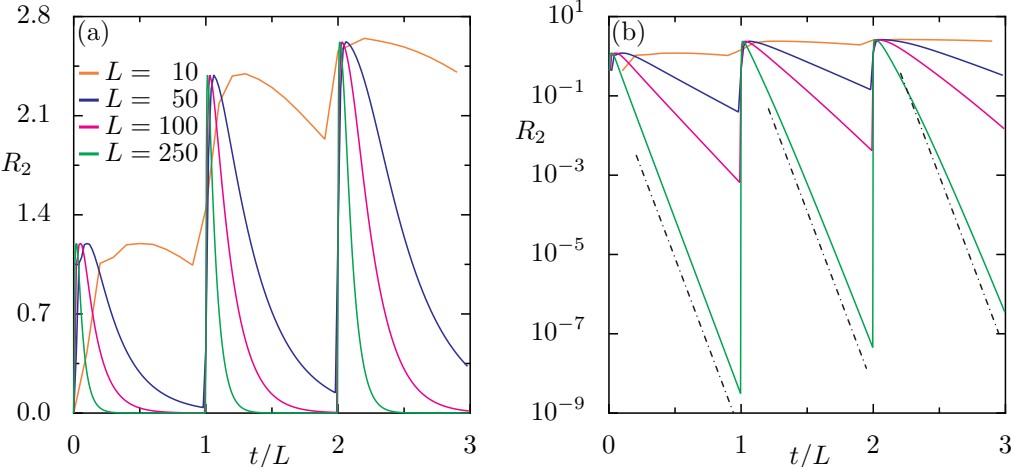

Figure 4: Second operator Rényi entropy for $q = 2$ and $l = 2$ for a generic impurity interaction, $a$ being the spin-$z$ operator, for various system sizes in (a) linear and (b) semi-logarithmic scale. (b) The dash-dotted lines illustrates the asymptotic scaling $|\lambda_0|^{\delta}$.

The corresponding left eigenvector is again the rainbow state $|r_\tau\rangle \in \left(\mathbb{C}^{q^2}\right)^{\otimes 2\tau}$, which appropriately normalized now reads

$$|r_\tau\rangle = q^{-\tau} \sum_{\alpha_1,\dots,\alpha_\tau = 0}^{q-1} |\alpha_1 \alpha_2 \cdots \alpha_\tau \alpha_\tau \cdots \alpha_2 \alpha_1\rangle \tag{68}$$

$$= q^{-\tau} \underbrace{\left\lfloor \; \underbrace{\lfloor \; \lfloor \sqcup \rfloor \; \rfloor}\; \right\rfloor}_{\tau} \tag{69}$$

The projection onto the eigenspace corresponding to eigenvalue 1 is then given by $\mathcal{P}_\tau = q^\tau |\circ \circ \cdots \circ\rangle \langle r_\tau|$ with the prefactor ensuring proper normalization of the left and right eigenvector and hence $\mathcal{P}_\tau^2 = \mathcal{P}_\tau$.

In the context of operator entanglement we call a generic impurity interaction completely chaotic, if there is no additional linear independent eigenvector for a unimodular eigenvalue of $\mathcal{T}_\tau$ for any $\tau$ and when there is a finite spectral gap between eigenvalue 1 and the subleading eigenvalue $\lambda_0$. Numerics suggest that this is the generic situation; see App. B. Repeating the same arguments from Sec. (3.2.1) by replacing $q$ by $q^2$ in intermediate steps, yields the reduced density matrix described by Eq. (40) and the entanglement entropies (41). For various system sizes $L$ we obtain the second Rényi entropy also numerically by contracting the tensor network (65) and depict it in Fig. 4 for subsystem size $l + 1 = 3$. There the asymptotic exponential dependence $|\lambda_0|^{\delta}$ is well confirmed for the largest system size (dashed line in (b)) and holds even for moderately large systems $L > 50$. Similar to the entanglement dynamics of states from gates which support a vacuum state, Rényi entropies of higher order $n > 2$ (not shown) agree with the $n = 2$ case.

### 4.2.2 T-dual impurity interactions

For the case of T-dual impurity interactions, the entanglement dynamics for local traceless operators acting non-trivially at the boundary can also be treated exactly for large systems and large subsystems. That is, in the limit $L, t, l \to \infty$, when limits are taken in the order described in Sec. 3.2.2. However, unlike the previous section, the leading part of the spectrum

of transfer matrices for folded T-dual impurity interactions is different from the ones studied in Sec. 3.2.2.

**Spectrum of transfer matrices:** The main difference to the case of initial product states lies in the unitality and dual unitality of the folded gate $V = WS$ in addition to dual unitarity of the folded gates. This gives rise to additional eigenvectors of $\mathcal{T}_\tau$ for leading eigenvalue 1. Unitality is a general property of the folded gates and was introduced in Sec. 4.2.1 already. On the other hand, dual unitality of the dual folded gate $\tilde{V}$ is defined akin to the state setting

$$\tilde{V}\,|\circ\circ\rangle = |\circ\circ\rangle\,, \qquad (\tilde{V}^\dagger)\,|\circ\circ\rangle = |\circ\circ\rangle\,, \tag{70}$$

and similarly for $\langle\circ\circ|$. This can be diagrammatically depicted as

$$\tag{71}$$

These properties give rise to $\tau+1$ linear independent eigenvectors of $\mathcal{T}_\tau$ for eigenvalue 1 given by [62, 75]

$$|s_x\rangle = |\circ\rangle^{\otimes \tau-x} \otimes |r_x\rangle \otimes |\circ\rangle^{\otimes \tau-x} \in \left(\mathbb{C}^{q^2}\right)^{\otimes 2\tau}\,, \tag{72}$$

constructed from the rainbow states, Eq. (33), $|r_x\rangle$ for $x \in \{1, \ldots, \tau\}$ and $|\circ\rangle^{\otimes 2\tau}$. In this case, we call the impurity completely chaotic if there are no other linearly independent eigenvectors with unimodular eigenvalue. In what follows, we first consider $l > 0$ and $\tau > 0$ in order to avoid constraints arising from the small size of the tensor networks. We shall discuss the other cases separately later.

One thing to immediately note about the state $|s_x\rangle$ is that they are not orthonormal, as $\langle s_x|s_y\rangle = q^{-|x-y|}$. Hence, we need to apply the Gram-Schmidt procedure to obtain a orthonormal set of eigenvectors given by

$$|t_0\rangle = |\circ\rangle^{\otimes 2\tau}\,, \tag{73}$$

$$|t_x\rangle = \frac{q}{\sqrt{q^2-1}}\left(|s_x\rangle - \frac{1}{q}|s_{s-1}\rangle\right)\,, \quad \text{for } x \in \{1, \ldots, \tau\}\,. \tag{74}$$

Thus the projection onto the eigenvalue 1 eigenspace is $\mathcal{P}_\tau = \sum_{x=0}^{\tau} |t_x\rangle\langle t_x|$. Also note that for T-dual impurity interactions left and right eigenvectors for eigenvalue 1 coincide. In particular, as $|t_0\rangle$ is both a left and a right eigenvector, $\mathcal{T}_\tau$ is the vectorization of a unital CPTP map.

**Asymptotic reduced density matrix:** We now derive the asymptotic reduced density matrix as $L, t \to \infty$ in the non-resonant case and briefly comment on the resonant case later. The degenerate eigenspace for eigenvalue 1 gives rise to a slightly more complex structure of the reduced density matrix. Upon replacing the transfer matrices $\mathcal{T}_\tau^{L-\delta-l_1}$ by $\mathcal{P}_\tau$ only the term $|t_\tau\rangle\langle t_\tau|$ gives a non-vanishing contribution to the reduced density matrix. For all the other terms the leftmost tensor factor $\langle\circ|$ in $\langle t_x|$ allows for contracting the leftmost column of the tensor network (65) due to unitality of the folded gate $V$ and yields a factor of $\langle\circ|a\rangle = 0$. By the same argument only the first two of the four terms

$$|t_\tau\rangle\langle t_\tau| = \frac{q^2}{q^2-1}\left(|s_\tau\rangle\langle s_\tau| - \frac{1}{q}|s_{\tau-1}\rangle\langle s_\tau| - \frac{1}{q}|s_\tau\rangle\langle s_{\tau-1}| + \frac{1}{q^2}|s_{\tau-1}\rangle\langle s_{\tau-1}|\right) \tag{75}$$

give a non-vanishing contribution to the reduced density matrix. Hence, we can replace $\mathcal{T}_\tau$ by $\frac{q^2}{q^2-1}\left(|s_\tau\rangle\langle s_\tau| - \frac{1}{q}|s_{\tau-1}\rangle\langle s_\tau|\right)$ and similarly for $\mathcal{T}_{\tau+1}$. This yields the asymptotic reduced density matrix as

$$\rho_l(t) = \tilde{\rho}_l^{(\tau)}(t) - \frac{1}{q^2}\tilde{\rho}_l^{(\tau-1)}(t)\,, \tag{76}$$

where

$$\left(\tilde{\rho}_l^{(\tau)}(t)\right)_{\beta_0\cdots\beta_l}^{\alpha_0\cdots\alpha_l} = \frac{q}{q^2-1}\,\langle r_\tau|(\mathcal{A}_{\tau+1})_{\beta_0\cdots\beta_l}^{\alpha_0\cdots\alpha_l}|r_{\tau+1}\rangle\,. \tag{77}$$

This can be diagrammatically represented as

$$\tilde{\rho}_l^{(\tau)}(t) = \frac{1}{q^2-1}\frac{1}{q^{2\tau}}$$ 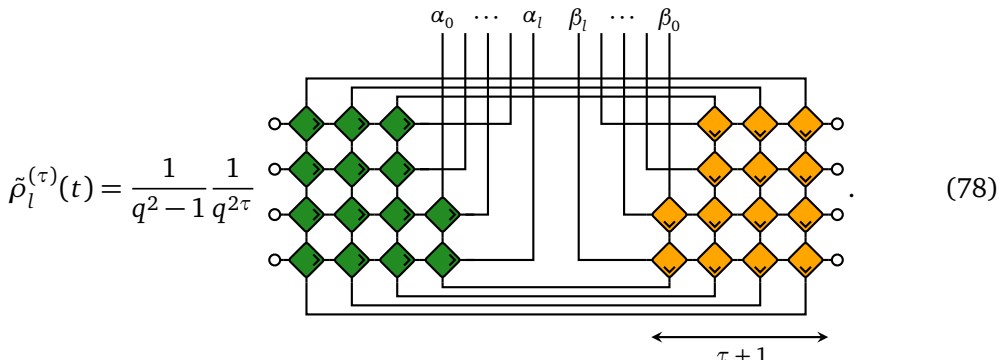 $$. \tag{78}$$

Unfortunately, Eq. (76) can not be simplified further except of $l = 0$ as was the case for states, which we shall discuss separately in 4.2.2.

**Rényi entropies for large subsystems $l \to \infty$:**   Now we shall derive the asymptotics of the Rényi entropies when also the subsystem is large, i.e. we take the limit $l \to \infty$ in a similar manner as in the case of states. As the asymptotic reduced density matrix, Eq. (76), is the difference of two terms, the computation is more involved than in the case of states. We moreover restrict ourselves to $\tau > 0$ in order to avoid additional complications due to small networks. We first sketch the main steps before getting into the details of the computation. There are four main steps we need to do to obtain our desired result.

1. **Rearranging** $\mathrm{tr}(\rho_l(t)^n)$**:** We rewrite $\mathrm{tr}(\rho_l(t)^n)$ as an alternating sum of terms of the form $\langle\boldsymbol{\sigma}|\mathcal{T}_{\sigma_n\sigma_{n-1}}^l \otimes \cdots \otimes \mathcal{T}_{\sigma_1\sigma_n}^l|\boldsymbol{\sigma}\rangle$ for suitable states $|\boldsymbol{\sigma}\rangle$, $\sigma_i \in \{\tau-1,\tau\}$ and generalized transfer matrices $\mathcal{T}_{\sigma_i\sigma_{i-1}}$ with similar spectral properties as the $\mathcal{T}_\tau$.

2. **Taking the limit $l \to \infty$:** Upon replacing the generalized transfer matrices by the projection onto their leading eigenvalue 1 for large $l$ most of the terms in the sum above cancel and we obtain $\mathrm{tr}(\rho_l(t)^n) \propto \langle\boldsymbol{\sigma}^\tau|\mathcal{P}_\tau^{\otimes n}|\boldsymbol{\sigma}^\tau\rangle - \langle\boldsymbol{\sigma}^{\tau-1}|\mathcal{P}_{\tau-1}^{\otimes n}|\boldsymbol{\sigma}^{\tau-1}\rangle$ with the states $|\boldsymbol{\sigma}^\tau\rangle$ similar as for states and the $\mathcal{P}_\tau$ as in the previous section.

3. **Evaluating matrix elements $\langle\boldsymbol{\sigma}^\tau|\mathcal{P}_\tau^{\otimes n}|\boldsymbol{\sigma}^\tau\rangle$:** Inserting $\mathcal{P}_\tau = \sum_x |t_x\rangle\langle t_x|$ in the first term all but the term $|t_\tau\rangle\langle t_\tau|$ are canceled by $\mathcal{P}_{\tau-1}$ in the second term and we are left with $\mathrm{tr}(\rho_l(t)^n) \propto \langle\boldsymbol{\sigma}^\tau|(|t_\tau\rangle\langle t_\tau|)^{\otimes n}|\boldsymbol{\sigma}^\tau\rangle$.

4. **Computing the overlap $\langle\boldsymbol{\sigma}^\tau|(|t_\tau\rangle)^{\otimes n}$:** Evaluating $\langle\boldsymbol{\sigma}^\tau|(|t_\tau\rangle\langle t_\tau|)^{\otimes n}|\boldsymbol{\sigma}^\tau\rangle$ eventually gives the final result in Eq. (109).

**1. Rearranging** $\mathrm{tr}(\rho_l(t)^n)$**:**   From Eq. (76) we obtain

$$\mathrm{tr}(\rho_l(t)^n) = \sum_{\boldsymbol{\sigma}\in\{\tau-1,\tau\}^n}\left(\frac{-1}{q^2}\right)^{\sharp\boldsymbol{\sigma}}\mathrm{tr}\left(\tilde{\rho}_l^{(\sigma_1)}(t)\tilde{\rho}_l^{(\sigma_2)}(t)\cdots\tilde{\rho}_l^{(\sigma_n)}(t)\right), \tag{79}$$

with the $\tilde{\rho}_l^{(\sigma)}$ defined in Eq. (77) and where we define $\sharp\boldsymbol{\sigma} := |\{i \in \{1,\ldots,n\} : \sigma_i = \tau-1\}|$. For subsequent calculations it is convenient to rewrite this in the form,

$$\mathrm{tr}(\rho_l(t)^n) = q^2\left(\frac{1}{q^2-1}\frac{1}{q^{2\tau}}\right)^n \sum_{\boldsymbol{\sigma}\in\{\tau-1,\tau\}^n}(-1)^{\sharp\boldsymbol{\sigma}}\mathrm{tr}\left(\hat{\rho}_l^{(\sigma_1)}(t)\hat{\rho}_l^{(\sigma_2)}(t)\cdots\hat{\rho}_l^{(\sigma_n)}(t)\right), \tag{80}$$

where the first factor $q^2$ arises similar as in the case of states by (repeatedly) contracting gates $V$ connected to the output legs $\alpha_0$ and $\alpha_2$ of the $i$-th replica with the adjoint gates $V^\dagger$ connected to the input legs $\beta_0$ and $\beta_2$ of the $(i-1)$-th replica. The second factor comes from the normalization constants (and the prefactor $1/q^2$) occuring in Eq. (78) (and Eq. (76)). Finally, $\hat{\rho}_l^{(\sigma_i)}(t)$ is given by the tensor network representation

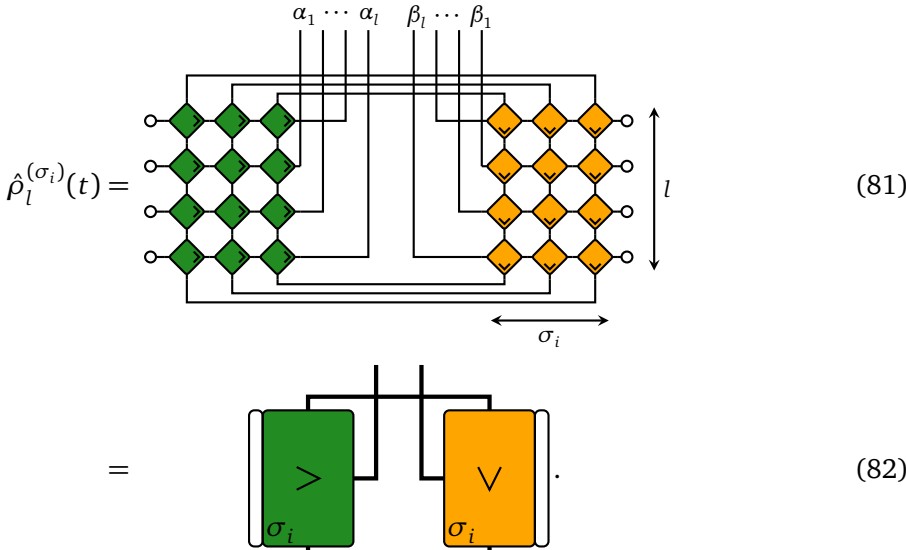

$$\hat{\rho}_l^{(\sigma_i)}(t) = \qquad\qquad\qquad\qquad\qquad\qquad \tag{81}$$

$$= \qquad\qquad\qquad\qquad\qquad . \tag{82}$$

This is similar as in the case of states but we now additionally indicate the size of the blocks in their respective bottom left corner. This also fixes the dimension of the Hilbert spaces carried by the wires connecting forward and backward block to be $q^{2\sigma_i}$. Again the above simplification implies that our subsequent results are independent from $\delta$.

For fixed $\boldsymbol{\sigma} \in \{\tau-1, \tau\}^n$ we can repeat the argument from Sec. 3.2.2 to obtain a similar equation as Eq. (52) by reshuffling the forward and backward parts of subsequent replicas. The resulting tensor networks have the same structure but in the operator case the wires connecting the (reshuffled) replica $i$ with replica $i-1$ carry Hilbert spaces whose dimension $q^{2\sigma_i}$ depends on the index $i$ of the replica. The (reshuffled) replicas can now be described in terms of the generalized transfer matrices $\mathcal{T}_{\sigma_i, \sigma_{i-1}}$ acting on $\left(\mathbb{C}^{q^2}\right)^{\otimes\sigma_i} \otimes \left(\mathbb{C}^{q^2}\right)^{\otimes\sigma_{i-1}}$ which are defined by their diagrammatic representation

$$\mathcal{T}_{\sigma_i, \sigma_{i-1}} = \qquad\qquad\qquad\qquad \tag{83}$$

shown here for $\sigma_i = \tau = 3$ and $\sigma_{i-1} = \tau-1 = 2$. In particular, one has $\mathcal{T}_{\tau\tau} = \mathcal{T}_\tau$. The replicas in the operator version of Eq. (52) can now be written as

$$\mathcal{T}_{\sigma_i \sigma_{i-1}}^l = \qquad\qquad\qquad\qquad \tag{84}$$

To complete the reshuffling of replicas leading to the operator version of Eq. (52) we introduce vectorized operators $|\boldsymbol{\sigma}\rangle$ which connect the replicas. Again, the state $|\boldsymbol{\sigma}\rangle$ will be obtained from the $n$-fold tensor product of rainbow states shifted by "half a replica". However, the individual factors now are states in $\left(\mathbb{C}^{q^2}\right)^{\otimes 2\sigma_i}$ and hence depend on the index $i$ of the replicas

they connect. To make the above precise we first define $|\boldsymbol{\sigma}| = 2\sum_i \sigma_i$. We again denote by $\eta_{-\sigma_n} \in S_{|\boldsymbol{\sigma}|}$ the $|\boldsymbol{\sigma}|$ periodic shift by $-\sigma_n$ and by $\mathbb{P}$ the unitary representation of $S_{|\boldsymbol{\sigma}|}$ which permutes the tensor factors in $\left(\mathbb{C}^{q^2}\right)^{\otimes|\boldsymbol{\sigma}|}$. We then define $|\boldsymbol{\sigma}\rangle \in \left(\mathbb{C}^{q^2}\right)^{\otimes|\boldsymbol{\sigma}|}$ by

$$|\boldsymbol{\sigma}\rangle = q^{\frac{|\boldsymbol{\sigma}|}{2}} \mathbb{P}_{\eta_{-\sigma_n}} \left(|r_{\sigma_n}\rangle \otimes |r_{\sigma_{n-1}}\rangle \otimes \cdots \otimes |r_{\sigma_1}\rangle\right) \tag{85}$$

$$= \boxed{\;\underline{\;\;\rule{0pt}{8pt}\;\;}\quad\underline{\;\;\rule{0pt}{8pt}\;\;}\quad\underline{\;\;\rule{0pt}{8pt}\;\;}\;\cdots\;\underline{\;\;\rule{0pt}{8pt}\;\;}\;}. \tag{86}$$

In the above network the wire reaching from left to right carries the Hilbert space $\left(\mathbb{C}^{q^2}\right)^{\otimes\sigma_n}$ of dimension $q^{2\sigma_n}$ and the inner wires carry Hilbert spaces of dimensions $d = q^{2\sigma_{n-1}}, q^{2\sigma_{n-2}}, \ldots, q^{2\sigma_1}$ (left to right). In particular for the case, where all the $\sigma_i$ are the same, i.e., for

$$\boldsymbol{\sigma}^{\tau} = (\tau, \tau, \ldots, \tau) \quad \text{and} \quad \boldsymbol{\sigma}^{\tau-1} = (\tau-1, \tau-1, \ldots, \tau-1) \tag{87}$$

we obtain the analog of the states defined in Sec. 3.2.2. Finally we arrive at

$$\mathrm{tr}\left(\hat{\rho}_l^{(\sigma_1)}(t)\hat{\rho}_l^{(\sigma_2)}(t)\cdots\hat{\rho}_l^{(\sigma_n)}(t)\right) = \langle\boldsymbol{\sigma}|\, \mathcal{T}_{\sigma_n\sigma_{n-1}}^l \otimes \mathcal{T}_{\sigma_{n-1}\sigma_{n-2}}^l \otimes \cdots \otimes \mathcal{T}_{\sigma_1\sigma_n}^l \,|\boldsymbol{\sigma}\rangle \,. \tag{88}$$

This concludes the first step. The above expression can be only evaluated further in the limit $l \to \infty$.

**2. Taking the limit $l \to \infty$:** As the generalized transfer matrices enter to the power of $l$, the above expression can be evaluated in the limit $l \to \infty$ by replacing the $\mathcal{T}_{\sigma_i\sigma_{i-1}}$ by their leading eigenvalue and the projection onto the corresponding eigenspace.

The $\mathcal{T}_{\sigma_i\sigma_{i-1}}$ are non-expanding, unital, CPTP maps with leading eigenvalue 1. Unitality and (dual) unitarity of the gate $V$ give rise to $\min\{\sigma_{i-1}, \sigma_i\} + 1$ linearly independent eigenvectors. For the completely chaotic T-dual impurity interactions considered here, these are the only eigenvectors, since one has $\mathrm{spec}\left(\mathcal{T}_{\sigma_1,\sigma_2}\right) \subseteq \mathrm{spec}(\mathcal{T}_\tau)$ for any $\tau \geq \max\{\sigma_1, \sigma_2\}$ as a consequence of (dual) unitarity. More precisely, given a right eigenvector $|\lambda\rangle$ of $\mathcal{T}_{\sigma_1,\sigma_2}$ with eigenvalue $\lambda$ the vector $|\circ\rangle^{\otimes\sigma_1-\tau} \otimes |\lambda\rangle \otimes |\circ\rangle^{\otimes\sigma_2-\tau}$ is an eigenvector of $\mathcal{T}_\tau$ with the same eigenvalue. Adapting this argument to the eigenvalue 1 for completely chaotic impurity interactions the projections $\mathcal{P}_{\sigma_i,\sigma_{i-1}}$ onto the corresponding eigenspace are given by

$$\mathcal{P}_{\tau\tau} = \mathcal{P}_\tau\,, \tag{89}$$

$$\mathcal{P}_{\tau\tau-1} = |\circ\rangle\langle\circ| \otimes \mathcal{P}_{\tau-1}\,, \tag{90}$$

$$\mathcal{P}_{\tau-1\tau} = \mathcal{P}_{\tau-1} \otimes |\circ\rangle\langle\circ|\,, \tag{91}$$

with $\mathcal{P}_\tau$ the corresponding projection for $\mathcal{T}_\tau$ introduced above. Hence,

$$\mathrm{tr}\left(\hat{\rho}_l^{(\sigma_1)}(t)\hat{\rho}_l^{(\sigma_2)}(t)\cdots\hat{\rho}_l^{(\sigma_n)}(t)\right) = \langle\boldsymbol{\sigma}|\, \mathcal{P}_{\sigma_n\sigma_{n-1}} \otimes \mathcal{P}_{\sigma_{n-1}\sigma_{n-2}} \otimes \cdots \otimes \mathcal{P}_{\sigma_1\sigma_n} \,|\boldsymbol{\sigma}\rangle \,, \tag{92}$$

up to terms exponentially suppressed with $l$.

The above expression is equal for all $\boldsymbol{\sigma} \neq \boldsymbol{\sigma}^\tau$ and hence in particular equals the expression for $\boldsymbol{\sigma}^{\tau-1}$. To see this, first consider $\boldsymbol{\sigma} \in \{\tau-1, \tau\}^n$ with not all entries identical. Thus there is $j \in \{1, \ldots, n\}$ with $\sigma_j = \tau - 1$ and $\sigma_{j-1} = \tau$. A straightforward computation then shows that contracting the projection $\mathcal{P}_{\tau\sigma_{j-2}}$ with $\langle\circ|$ and $|\circ\rangle$ on the left yields the projection $\mathcal{P}_{\tau-1\sigma_{j-2}}$ acting on a smaller space. Formally, this reads

$$\left(\langle\circ| \otimes \mathbb{1}\right)\mathcal{P}_{\tau\sigma_{j-2}}\left(|\circ\rangle \otimes \mathbb{1}\right) = \mathcal{P}_{\tau-1\sigma_{j-2}}\,, \tag{93}$$

where $\mathbb{1}$ denotes the identity on $\left(\mathbb{C}^{q^2}\right)^{\otimes \tau - 1 + \sigma_{j-2}}$. This is obvious for $\sigma_{j-2} = \tau - 1$ and for $\sigma_{j-2} = \tau$ follows from writing

$$\mathcal{P}_{\tau\tau} = \mathcal{P}_\tau = |t_\tau\rangle\langle t_\tau| + |\circ\rangle\langle\circ| \otimes \mathcal{P}_{\tau-1} \otimes |\circ\rangle\langle\circ| \,, \tag{94}$$

and noting that

$$(\langle\circ| \otimes \mathbb{1})|t_\tau\rangle\langle t_\tau|(|\circ\rangle \otimes \mathbb{1}) = 0 \,. \tag{95}$$

From the above properties it follows that

$$\langle\boldsymbol{\sigma}|\mathcal{P}_{\sigma_n\sigma_{n-1}} \otimes \mathcal{P}_{\sigma_{n-1}\sigma_{n-2}} \otimes \cdots \otimes \mathcal{P}_{\sigma_1\sigma_n}|\boldsymbol{\sigma}\rangle = \langle\boldsymbol{\pi}|\mathcal{P}_{\pi_n\pi_{n-1}} \otimes \mathcal{P}_{\pi_{n-1}\pi_{n-2}} \otimes \cdots \otimes \mathcal{P}_{\pi_1\pi_n}|\boldsymbol{\pi}\rangle \,,$$

if $\pi_i = \sigma_i$ for $i \neq j - 1$ and $\pi_{j-1} = \tau - 1$.

This argument is best illustrated diagrammatically by

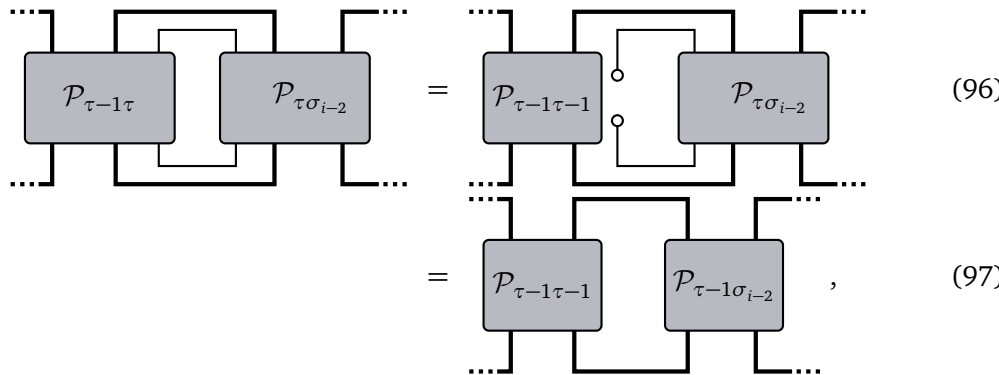

where the gray boxes represent the indicated projections with which we replaced $\mathcal{T}_{\sigma_j\sigma_{j-1}}$ in Eq. (109). At the extreme left, we have $\mathcal{P}_{\sigma_j\sigma_{j-1}} = \mathcal{P}_{\tau-1\tau}$. The wires to the left (right) carry the Hilbert space $\mathbb{C}^d$ of dimension $d = q^{2(\tau-1)}$ ($d = q^{2(\sigma_{j-2})}$) and for the central thick wires $d = q^{2(\tau-1)}$, while for the central thin wires $d = q^2$. By repeated use of the above argument it follows that $\langle\boldsymbol{\sigma}|\mathcal{P}_{\sigma_n\sigma_{n-1}} \otimes \mathcal{P}_{\sigma_{n-1}\sigma_{n-2}} \otimes \cdots \otimes \mathcal{P}_{\sigma_1\sigma_n}|\boldsymbol{\sigma}\rangle = \langle\boldsymbol{\sigma}^{\tau-1}|\mathcal{P}_{\tau-1}^{\otimes n}|\boldsymbol{\sigma}^{\tau-1}\rangle$. Finally using $\sum_{\boldsymbol{\sigma}\neq\boldsymbol{\sigma}^\tau}(-1)^{\sharp\boldsymbol{\sigma}} = \sum_{k=1}^n \binom{n}{k}(-1)^k = -1$, as follows from the binomial theorem, we simplify Eq. (80) as

$$\mathrm{tr}(\rho_l(t)^n) = q^2 \left(\frac{1}{q^2-1}\frac{1}{q^{2\tau}}\right)^n \left(\langle\boldsymbol{\sigma}^\tau|\mathcal{P}_\tau^{\otimes n}|\boldsymbol{\sigma}^\tau\rangle - \langle\boldsymbol{\sigma}^{\tau-1}|\mathcal{P}_{\tau-1}^{\otimes n}|\boldsymbol{\sigma}^{\tau-1}\rangle\right) \,. \tag{98}$$

This concludes the second step.

**3. Evaluating matrix elements $\langle\boldsymbol{\sigma}^\tau|\mathcal{P}_\tau^{\otimes n}|\boldsymbol{\sigma}^\tau\rangle$:** Now, we shall show that the second term in Eq. (98) almost completely cancels the first term. To this end we first insert Eq. (94) into the first term. Then a similar argument as sketched in Eq. (97) yields

$$\langle\boldsymbol{\sigma}^\tau|\mathcal{P}_\tau^{\otimes n}|\boldsymbol{\sigma}^\tau\rangle = \langle\boldsymbol{\sigma}^\tau|(|t_\tau\rangle\langle t_\tau|)^{\otimes n}|\boldsymbol{\sigma}^\tau\rangle + \langle\boldsymbol{\sigma}^{\tau-1}|\mathcal{P}_{\tau-1}^{\otimes n}|\boldsymbol{\sigma}^{\tau-1}\rangle \,, \tag{99}$$

where mixed terms in the $n$-fold tensor product cancel due to Eq. (95). Clearly, the second term in the above equation is exactly canceled by the second term in Eq. (98). This yields

$$\mathrm{tr}(\rho_l(t)^n) = q^2 \left(\frac{1}{q^2-1}\frac{1}{q^{2\tau}}\right)^n \left|\langle\boldsymbol{\sigma}^\tau|\left(|t_\tau\rangle^{\otimes n}\right)\right|^2 \,. \tag{100}$$

**4. Computing the overlap** $\langle \sigma^\tau | (|t_\tau\rangle)^{\otimes n}$: Finally, we are left with the computation of the overlap $\left| \langle \sigma^\tau | (|t_\tau\rangle)^{\otimes n} \right|^2$. Using the fact, that $|t_\tau\rangle$ coincides with $|r_{\tau-1}\rangle$ on all but the leftmost and rightmost tensor factor the overlap factorizes as

$$\left| \langle \sigma^\tau | (|t_\tau\rangle)^{\otimes n} \right|^2 = \left| \langle \sigma^{\tau-1} | (|r_{\tau-1}\rangle)^{\otimes n} \right|^2 \left| \langle \sigma^1 | (|t_1\rangle)^{\otimes n} \right|^2 , \tag{101}$$

where in the last factor $|t_1\rangle = \frac{q}{\sqrt{q^2-1}} \left( |r_1\rangle - \frac{1}{q} |\circ\circ\rangle \right)$, i.e., the state $|t_\tau\rangle$ in Eq. (74) for $\tau = 1$. Using the diagrammatic representation of states, the first factor gives $q^{-(\tau-1)(2n-4)}$. Similarly, for the second factor we obtain

$$\langle \sigma^1 | (|t_1\rangle)^{\otimes n} = \left( \frac{q}{\sqrt{q^2-1}} \right)^n \left( \langle \sigma^1 | (|r_1\rangle)^{\otimes n} - \langle \sigma^1 | \left( \frac{1}{q} |\circ\circ\rangle \right)^{\otimes n} \right) = \left( q^2 - 1 \right)^{1-\frac{n}{2}} , \tag{102}$$

as all the mixed terms in the $n$-fold tensor product $(|t_1\rangle)^{\otimes n}$ cancel by a similar argument as for deriving Eq. (98). Combining everything we conclude the fourth step by obtaining

$$\left| \langle \sigma^\tau | (|t_\tau\rangle)^{\otimes n} \right|^2 = \left( \frac{q^2}{(q^2-1)q^{2\tau}} \right)^{n-2} . \tag{103}$$

This ultimately leads to

$$\mathrm{tr}(\rho_l(t)^n) = \left( \frac{q}{(q^2-1)q^{2\tau}} \right)^{2(n-1)} , \tag{104}$$

up to terms exponentially suppressed at least as $|\lambda_0|^l$. This gives the Rényi entropy as

$$R_n(t) = 2\tau \ln(q^2) - 2\ln\left( \frac{q}{q^2-1} \right) , \tag{105}$$

independent from $n$ up to terms which vanish as $l \to \infty$. For the case $\tau = 0$, applying the above line of reasoning gives

$$\mathrm{tr}(\rho_l(t)^n) = \left( \frac{1}{(q^2-1)} \right)^{n-1} , \tag{106}$$

as the exact result even for finite $l$. This corresponds to the Rényi entropy

$$R_n(t) = \ln(q^2 - 1) . \tag{107}$$

However, originating from the subleading terms of the asymptotic reduced density matrix Eq. (76) the subleading terms of the entropies scale as $|\lambda_0|^\delta$ and hence give rise to non-trivial initial dynamics. Finally, for completeness, we mention the corresponding result for the resonant case and $\tau > 1$, since the derivation is similar. We have

$$\mathrm{tr}(\rho_l(t)^n) = \left( \frac{q^2}{(q^2-1)q^{2\tau}} \right)^{2(n-1)} . \tag{108}$$

This gives the Rényi entropies as

$$R_n(t) = 2\tau \ln(q^2) - 2\ln\left( \frac{q^2}{q^2-1} \right) . \tag{109}$$

Even though the tensor network (65) allows for computing the reduced density matrix for large system size $L$ and small subsystem size $l$, direct numerical simulation fails for large $l$ as the

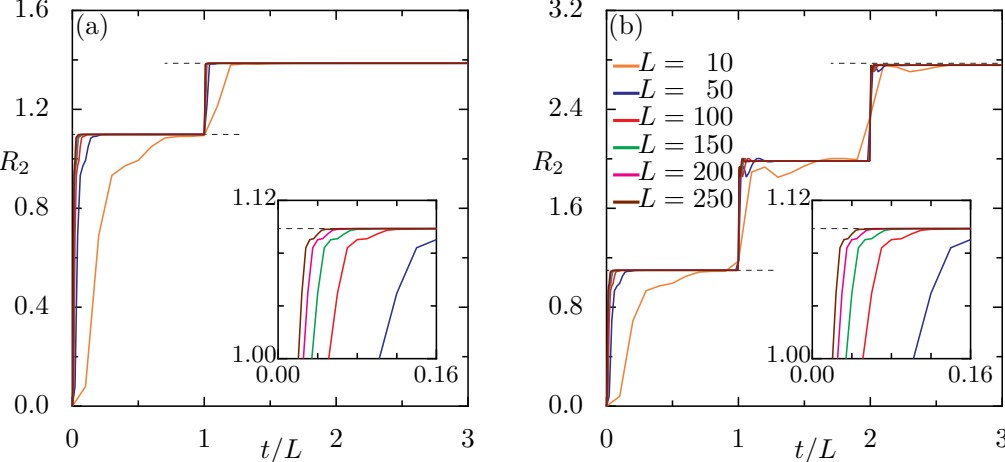

Figure 5: Second Rényi entropy for $q = 2$, a T-dual impurity interaction with $J = \pi/4 - 0.05$ in Eq. (42), and $a$ the spin-$z$ operator with (a) $l = 0$ and (b) $l = 1$ for various system sizes. The dashed lines corresponds to (a) Eq. (114) and (b) Eq. (107) as well as the maximum entropy $(l+1)\ln(q^2)$. The inserts show a magnification for initial times.

complexity of the computation grows exponentially with $l$. Nevertheless, at least the plateau-like structure suggested by Eq. (109), i.e., almost constant entanglement entropy for constant $\tau$, can be observed for small subsystem size as well. This is depicted in Fig. 5(b) for $l = 1$. Also the non-trivial initial dynamics as predicted by Eq. (107) is confirmed there; see inset. Moreover, we find the entanglement entropy to saturate at the maximum possible value after times $t = 2L$ ($\tau = 2$) for the example considered here. Additionally we numerically computed Rényi entropies of higher order $n > 2$ (not shown) and found them to coincide qualitatively with the results for $n = 2$. Similar to the entanglement dynamics of states for T-dual gates only the value at the plateaus for intermediate times, i.e, $\tau > 0$ but before entanglement entropies saturate, e.g., $\tau = 1$ in Fig. 5(b), depends very weakly on $n$. This indicates that even for the small numerically accessible subsystems sizes $l$ the non-zero eigenvalues of the reduced density matrix almost coincide as it is the case in the $l \to \infty$ limit.

**Rényi entropies for small subsystems $l = 0$:** As mentioned before, another case which allows for exact results is that of minimal subsystem size $l = 0$ for which the subsystem only contains the lattice site 0 at the boundary, whereas $L, t \to \infty$. In this situation we can obtain an exact expression of the reduced density matrix as described below.

Firstly, for $\tau \geq 1$ the asymptotic analysis from discussion earlier in this section applies for the non-resonant case. Hence, the asymptotic reduced density matrix is given by Eq. (76). Then we evaluate Eq. (77) further using $(\mathcal{A}_{\tau+1})^{\alpha}_{\beta} |r_{\tau+1}\rangle = q^{-1}\delta_{\alpha,\beta} |r_{\tau}\rangle$ to get

$$\left(\rho_l^{(\tau)}(t)\right)^{\alpha}_{\beta} = \frac{1}{q^2 - 1}\delta_{\alpha,\beta} \,. \tag{110}$$

Thus, we obtain $(\rho_0)^{\alpha}_{\beta}(t) = q^{-2}\delta_{\alpha,\beta}$ and hence

$$\rho_0(t) = \frac{1}{q^2}\mathbb{1}_{q^2} \tag{111}$$

is the infinite temperature state up to corrections proportional to $|\lambda_0|^L$.

Secondly, for $\tau = 0$, i.e., $0 \leq t < L$ the reduced density matrix takes the simple form

$$(\rho_0)^\alpha_\beta(t) = (\mathcal{A}_1)^\alpha_\beta \, \mathcal{T}_1^{\delta-1} |a\rangle \otimes |a\rangle \,. \tag{112}$$

For large $\delta \gg 1$ the transfer matrix $\mathcal{T}_1$ can again be replaced by the projection onto the eigenspace for the eigenvalue 1. Following the argument for general $l$ we see that only the terms proportional to $|r_1\rangle\langle r_1|$ and $|\circ\circ\rangle\langle r_1|$ give a non-vanishing contribution. The first term gives a contribution $\propto \mathbb{1}_{q^2}$ whereas the second term gives a contribution $\propto |\circ\rangle\langle\circ|$. Collecting both terms we obtain

$$\rho_0(t) = \frac{1}{q^2-1}\left(\mathbb{1}_{q^2} - |\circ\rangle\langle\circ|\right), \tag{113}$$

which corresponds to the infinite temperature state restricted to the subspace orthogonal to $|\circ\rangle$, i.e., of traceless operators. Consequently the corresponding Rényi entropies read

$$R_n(t) = \begin{cases} \ln\left(q^2-1\right), & \text{if } \tau = 0\,, \\ \ln\left(q^2\right), & \text{if } \tau > 0\,, \end{cases} \tag{114}$$

and are independent of $n$. For $\tau = 0$ this coincides with Eq. (106) and gives rise to the same non-trivial initial entanglement dynamics discussed there. In the resonant case, the same results can be obtained as in the non-resonant case, only for $t = L$, i.e., $\tau = 1$ and $\delta = 0$ the reduced density matrix and the corresponding entropies correspond to Eq. (114). In Fig. 5(a) we depict the second Rényi entropy for various system sizes obtained from contracting the tensor network (65) for $l = 0$. The asymptotic form of the entropies is approached fast even for moderately large system sizes. In the inset we additionally show the non-trivial initial entanglement dynamics.

## 5 Conclusion

We study the entanglement dynamics for both product states and local operators in a minimal model of many-body quantum chaos built from a locally perturbed free quantum circuit. Using a minimal but exact description of time evolution resulting from analytically integrating out the free part of the circuit we obtain tensor network representations of the reduced density matrices. We contract the tensor networks using a transfer matrix approach in spatial direction resulting in a simple form of the reduced density matrices in the limit of large system size $L$.

Then, depending on the choice of the perturbation, i.e., the impurity interaction at the system's boundary, we either compute the reduced density matrix or the corresponding Rényi entropies exactly. For the gates which exhibit a local vacuum state, the reduced density matrix of an initial product state is close to a pure and hence unentangled state at most times. Similar dynamics is observed in the reduced super density matrix of initially local operators for generic impurity interactions. It is only for times $t \approx \tau L$ in resonance with system size, that entanglement entropies are large in both cases. This results in untypical entanglement dynamics, of periodically spiking entanglement entropies, despite the system being chaotic in the sense of spectral statistics. In such chaotic systems entanglement entropies generically grow linearly in time. Hence our setting resembles an example where different notions of many-body quantum chaos, namely random-matrix like spectral fluctuations and linear growth of entanglement entropies do not coincide, as it is also the case when studying thermalization in the present setting [65].

In contrast we recover the entanglement dynamics of typical chaotic systems, i.e., linear growth of entanglement entropies, for T-dual impurity interactions when the size of the subsystem is large. This is the case both for initial product states and local operators. More precisely

entanglement grows linearly with $\tau$ leading to plateaus in the Rényi entropies in between resonant times. The height of the plateaus grows at maximum speed given by $2\log(d)$, with $d = q$ or $q^2$, respectively. Hence, as $\tau \approx t/L$, the speed of entanglement growth is reduced by a factor of $1/L$ compared to the maximum value, which we attribute to only one gate, i.e., the impurity interaction, of the in total $L$ gates of the circuit being entangling. One therefore might conjecture, that for a number of $n$ entangling gates one should get a correction $n/L$ to the maximal possible speed and that the maximum speed is recovered in the spatially homogeneous setting.

Our work hence provides an exact description of the entanglement dynamics in large systems for either arbitrary subsystem size (entanglement of states for gates with vacuum state and operator entanglement for generic gates) or in the limit of infinite subsystem size (T-dual gates). In the latter case our results explain the entanglement dynamics qualitatively even for small subsystems. However, for finite subsystems we are currently not able to address the question of saturation of entanglement entropies at late times. This is due to exponential scaling of the size of transfer matrices with $\tau$, which renders the large $\tau$ regime intractable via numerics. Unfortunately, this cannot be computed analytically as well due to the fact that, for finite subsystems the subleading part of the spectrum of the transfer matrices also becomes relevant, for which we lack an analytical description.

Hence, to address the question of saturation, one requires different techniques, e.g., methods based on a dual space-time swapped interpretation as recently introduced in Ref. [30], which is beyond the scope of this work. Also, if one was able to approach longer times, one might be able to study the phenomenon of entanglement barriers for operator entanglement in the boundary-chaos setting.

# Acknowledgements

FF thanks K. Klobas and B. Bertini for insightful discussions.

**Funding information** FF would like to acknowledge support from Deutsche Forschungsgemeinschaft (DFG) Project No. 453812159. RG would also like to acknowledge support from Grant No. J1-1698 from the Slovenian Research Agency (ARRS) and UKRI grant 'Non-ergodic quantum manipulation' EP/R029075/1. TP acknowledges support from research Program P1-0402 of ARRS.

# A    Spectral statistics of the circuit

In this appendix we present the level spacing distribution $p(s)$ for the boundary chaos circuit for the three different classes of impurity interactions – gates preserving a vacuum state, T-dual gates, and generic gates. The scaled level spacing $s_i = \frac{q^{L+1}}{2\pi}(\epsilon_{i+1} - \epsilon_i)$ is given by the difference of consecutive eigenphases/quasi-energies of the boundary chaos circuit $\mathcal{U}$ and is normalized to unit mean spacing. For the impurity interaction with a vacuum-preserving gate used in Fig.1 we depict $p(s)$ in Fig. 6(a), whereas (b) shows $p(s)$ for the T-dual impurity interaction from Fig. 5 and (c) shows $p(s)$ for the generic impurity interaction from Fig. 4. Each agrees well with the random matrix result for the respective symmetry class. For the T-dual case this is the circular orthogonal ensemble (COE) for $q = 2$ and the circular unitary ensemble (CUE) for larger $q$ (not shown). The other two cases correspond to the CUE for any $q$. The impurity interactions used for Fig. 2 and Fig. 3 yield similar level spacing distributions and are not shown separately. The correspondence between the level spacing distribution for the boundary chaos circuit and the respective random matrix results clearly indicates our setting

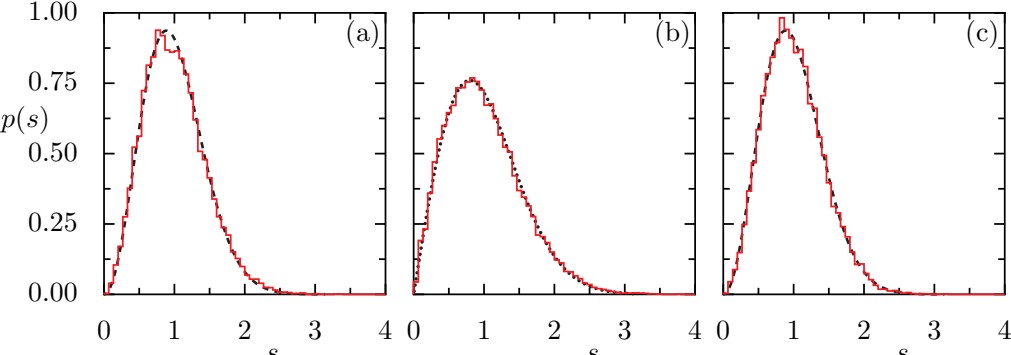

Figure 6: Level spacing distribution $p(s)$ for (a) impurity interaction with vacuum, (b) T-dual and (c) generic impurity interactions for $L + 1 = 14$ and $q = 2$. Dashed and dotted black lines correspond to the corresponding distribution for the CUE (a,c) and COE (b) respectively.

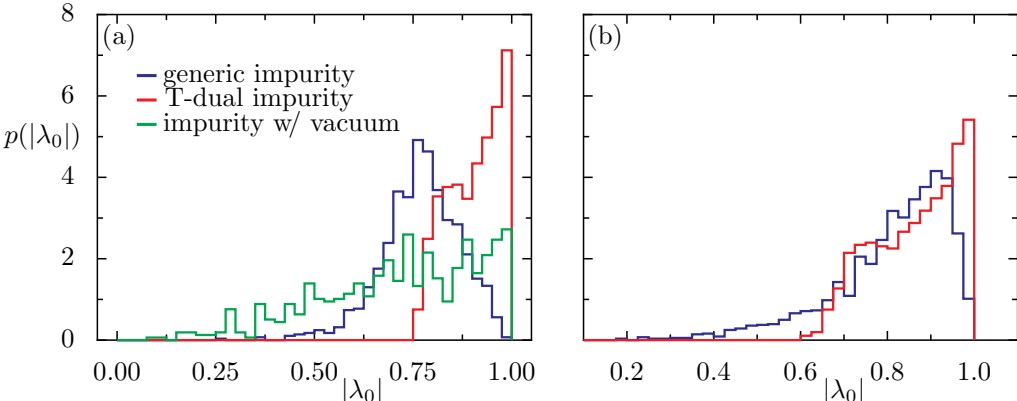

Figure 7: Distribution $p(|\lambda_0|)$ of the subleading eigenvalue $\lambda_0$ for (a) states with $\tau = 9$ and (b) operators with $\tau = 5$ from 10000 realizations from different classes of impurity interactions (see legend).

indeed leads to chaotic quantum systems in the sense of spectral statistics.

## B  Subleading eigenvalues of transfer matrices

Our analysis of entanglement dynamics, both for states and operators, requires subleading eigenvalues $\lambda$ of the transfer matrices $\mathcal{T}_\tau$ to be gapped from one, i.e., $|\lambda| < 1$. As this is out of scope of a rigorous proof we resort to extensive numerical studies to confirm this claim. Using Arnoldi iteration in the subspace orthorgonal to the eigenspace of the leading eigenvlaue 1 we compute the subleading eigenvalue of the transfer matrices for the largest accessible values of $\tau$. Note that $\mathcal{T}_\tau$ is a non-Hermitian (and in general non-normal) matrix of dimension $q^{2\tau}$ in the case of states, whereas it is of dimension $q^{4\tau}$ in the case of operators. We compute the subleading eigenvalue at size $\tau = 9$ for states and $\tau = 5$ for operators for 10000 realizations for qubits $q = 2$ for the different classes of impurity interactions. Here, we sample the generic gates Haar random from U(4), while we choose $u$ in the gate a with vacuum state, $U = 1 \oplus u$, Haar random from U(3). For T-dual gates we fix the interaction $J = \pi/4 - 0.05$ in Eq. (42) and choose the local unitaries $u_\pm, v_\pm$ Haar random from U(2). In Fig. 7(a) we show the distribution of the modulus of the subleading eigenvalue $|\lambda_0|$ for the case of states. For generic impurity

interactions we find the probabiltiy to dropp towards zero when $|\lambda_0|$ approaches 1 indicating a finite gap for random choices of the gate. In contrast, both for T-dual impurity interactions and those with a vacuum state we find the probability to be largest around 1 indictating finite probability to find arbitrary large subleading eigenvalue Nevertheless, we do not find a single instance where the subleading eigenvalue actually has modulus one and hence there will be at least a small gap for generic choices of the impurity interaction from these classes. A small spectral gap only implies, that the limiting entanglement dynamics for $L \to \infty$ is approached much slower. In Fig. 7(b) we additionally show the same data for the transfer matrices from the operator case and find qualitatively very similar behavior as for states.

## C  Impurity interactions for numerical computations

In this section we provide the impurity interactions $U$, entering Eq. (7), which we use for numeical computations. For the entanglement dynamics of states, presented in Sec. 3.2.1 from impurities which support a vacuum state the gate is

$$U = \begin{pmatrix} 1 & 0 & 0 & 0 \\ 0 & 0.56078693+i0.13052803 & -0.31583062-i0.08879493 & 0.59273587+i0.45772385 \\ 0 & 0.7123732+i0.2419316 & 0.39227097-i0.22401521 & -0.2246578-i0.42363082 \\ 0 & 0.30203025+i0.10607406 & -0.38000351+i0.7374988 & -0.45253113+i0.06659165 \end{pmatrix}. \tag{C.1}$$

For T-dual impurity interactions, discussed in Sec. 3.2.2, our choice of local unitaries in Eq. (42) as well as $J = \pi/4 - 0.05$ yields

$$U = \begin{pmatrix} -0.56511125+i0.14546062 & -0.34221162+i0.55985625 & 0.15649404-i0.42756842 & 0.13597384+i0.05611289 \\ -0.44338093+i0.48186603 & -0.0147065-i0.58867791 & 0.0635994-i0.13913641 & -0.39303688-i0.21582098 \\ -0.28160319+i0.15256069 & -0.18672499+i0.30249188 & -0.17678486+i0.82765589 & -0.18242321-i0.14666947 \\ -0.21884669+i0.28326661 & 0.04262872-i0.30742391 & -0.02241818+i0.22917526 & 0.72147776+i0.44942792 \end{pmatrix}. \tag{C.2}$$

In the case of generic impurity interactions from Sec. 3.2.3 we use

$$U = \begin{pmatrix} 0.36435611+i0.30859449 & 0.25372067+i0.09374407 & 0.55892459+i0.07675011 & 0.55887898-i0.26118756 \\ 0.38573468-i0.52730716 & -0.18706112+i0.16487939 & 0.3447626+i0.55123777 & -0.28602683+i0.08026939 \\ -0.02131933+i0.46880636 & 0.08412532-i0.47063255 & 0.44362124+i0.0356644 & -0.56044703+i0.19753834 \\ 0.1571885+i0.31658781 & 0.05215964+i0.79584427 & 0.00758074-i0.24669653 & -0.42106545-i0.0276125 \end{pmatrix}. \tag{C.3}$$

For the entanglement dynamics of operators and generic impurity interactions, discussed in Sec. 4.2.1 we choose the impurity interaction

$$U = \begin{pmatrix} -0.15302565-i0.00702436 & 0.07406427+i0.68998362 & -0.61881981-i0.11039248 & 0.01091446+i0.31579631 \\ -0.65851585-i0.32242472 & 0.10537759-i0.29765891 & -0.18430557+i0.16641164 & -0.54844977+i0.01534251 \\ -0.38846017-i0.12440906 & -0.25436805-i0.31744661 & -0.32211766-i0.15547394 & 0.71989747-i0.1481935 \\ 0.01525248-i0.52184426 & -0.42302265+i0.27259531 & 0.24344719+i0.59667051 & 0.16349771+i0.17937624 \end{pmatrix}, \tag{C.4}$$

whereas for the T-dual case, presented in Sec. 4.2.2, our choice of local unitaries in Eq. (42) as well as $J = \pi/4 - 0.05$ results in

$$U = \begin{pmatrix} -0.3923746-i0.30245775 & 0.32529251+i0.04037993 & 0.34083335+i0.45827838 & 0.47857571+i0.30314118 \\ -0.28166778+i0.31394062 & -0.00791122+i0.62361304 & -0.29015399-i0.40778701 & 0.30814909-i0.29616428 \\ -0.53469481-i0.02205057 & -0.3565329-i0.48333761 & -0.50931968-i0.09325641 & 0.11279488+i0.26843675 \\ -0.01098664+i0.53866553 & 0.28259919+i0.2510083 & -0.01856994-i0.39355537 & 0.10761974+i0.63248604 \end{pmatrix}. \tag{C.5}$$

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
