# Peer review of "Boundary Chaos: Exact Entanglement Dynamics"

_SciPost Physics, doi:SciPost Phys. 15, 092 (2023)_

## Round 2 · Referee Report · Pieter W. Claeys (Referee 1) · 2023-3-17

Strengths

1- Efficient tensor network representation of entanglement entropy in a minimal model of boundary chaos. 2- Exact results for (nontrivial) entanglement dynamics of both initial product states and initial local operators.

Weaknesses

1- Results are restricted to the very specific setup of boundary chaos. 2- Relatively short discussion of entanglement dynamics for initial product states with generic impurity.

Report

In this work the authors consider a minimal model for ergodic and mixing quantum many-body dynamics termed 'boundary chaos', where a brickwork quantum circuit consisting of non-interacting swap gates is perturbed by a single impurity interaction placed at the boundary. In a previous work by two of the authors it was shown that such a model is indeed chaotic and ergodic, in terms of both the spectral statistics and the dynamical correlations. Here the methods developed for this model are extended to the calculation of entanglement dynamics for initial product states and an initial local operator. The geometry of the setup already allows for an efficient tensor network representation of the entanglement entropy, such that numerical results can be obtained for large system sizes and long times. The authors then focus on various scenarios where exact analytics results can be obtained.

1) Initial product states: The authors consider two specific choices of impurity, leading to qualitatively different dynamics. If the impurity supports a 'vacuum' state, the entanglement dynamics exhibits periodic revivals at times proportional to the system size, and if the impurity is chosen to be T-dual the entanglement dynamics exhibit a staircase structure. For generic impurities no analytical treatment is possible, but numerical results suggest a mixture of these two scenarios.

2) Initial local operator: For generic impurities the operator entanglement dynamics is shown to behave in the same way as the entanglement dynamics of an initial product state with an impurity supporting a vacuum, i.e. exhibiting periodic revivals, due to a direct analogy in the calculation when moving to a 'vectorized' representation. For a T-dual impurity a staircase structure is again obtained.

These calculations are underlied by identifying a transfer matrix in the tensor network representation of the entanglement entropy and restricting to cases where the leading eigenspectrum of this transfer matrix can be analytically obtained, similar to various calculations in the literature on dual-unitary circuits.

The analytical calculations are clearly presented and convincing, and well supported by numerics. The paper is overall very well written and the results are sure to be of interest to the community working on entanglement dynamics and chaotic many-body systems. I am happy to recommend this paper for publication in SciPost Physics, provided some minor clarifying comments are addressed. The calculation of entanglement entropies in chaotic quantum many-body dynamics is a notoriously difficult problem, and the authors here present an exact solution for the entanglement dynamics in a minimal but nontrivial model. For this reason I believe that this paper satisfies the SciPost Physics acceptance criteria.

Requested changes

1- In Eq. (21) the definition of the permutation operator is somewhat ambiguous, and does not correspond to the previous definition (C5) from Ref. [65], which is referred to for more details. When trying to reproduce Eq. (22) for $L=5$ I find e.g. the inconsistent result that $\sigma_0(2) = 5 \textrm{ mod 5 }=\sigma_0(5) = 0 \textrm{ mod } 5$, but everything seems to be consistent when using the definition from Ref. [65]. Can the authors clarify?

2- When discussing the dynamics starting from an initial product state the authors consider the cases where the impurity either supports a vacuum state or is T-dual, leading to qualitatively different dynamics. Can the authors comment on what happens if the impurity supports both a vacuum state and is T-dual?

3- Section 3.2.3, presenting numerical results on the entanglement dynamics starting from an initial product state and generic impurity, is somewhat short and does not allow the reader to reproduce the conclusions. A single numerical result is given for a randomly chosen impurity and it is mentioned that the dynamics is a mix of the previously discussed cases, but it is not clear how generic/reproducable this behavior is. It would be useful to give the numerical parametrization of the impurity in Appendix to make the results reproducible. A brief discussion on how the observed dynamics depend on 'how close the impurity is to T-dual' could also be useful, but I do not insist on this.

4- On page 12, I assume that the transfer matrices correspond to the rows of the tensor network (25) rather than the columns, as is written. Is this correct?

5- Analytical results are obtained for Rényi entropies $R_n$ of arbitrary order $n$, but numerical comparisons are restricted to the second Rényi entropy $n=2$. Could the authors comment on how they would expect the numerics and the effect of the subleading terms in the analytical calculatoin to change for higher Rényi index?

6- There are some typos in the manuscript ('intergrable', 'impuirity','vecotrization', 'wit', 'dynamcis', 'sencond').

---

## Round 2 · Referee Report · Anonymous (Referee 2) · 2023-5-4

Strengths

  1. Computation of exact results for dynamics of entanglement in a many body system
  2. Extremely clear exposition of the diagrammatic methods upon which the calculations are based

Report

This paper presents interesting results on the dynamics of entanglement in a many qubit circuit in 1+1 dimensions, where the dynamics is only nontrivial at the boundary. Earlier work has established the apparent limited complexity of such a system, it has the spectral properties typical of quantum chaos. This work shows that these properties coexist with unusual entanglement dynamics, including revivals at regular intervals due to the simple dynamics of the bulk.

The work makes extensive use of tensor network diagrams, with most of the formal development begin carried out with some very intricate graphical manipulations. Fortunately these are explained very clearly, with the diagrams forming an integral part of the text. The developed formalism is able to deal with both state and operator entanglement measures on an almost identical footing.

I am happy to recommend acceptance, with only a few changes to request.

Requested changes

  1. A few minor typos "intergrable" (p3) "vecotrization" (p7)
  2. In the description of the diagrams Eq. (24) and (25) it might be useful to explicitly state the values of $L$, $l$, and $t$
  3. In Eq. (40) and after it might be clearer to state that the eigenvalue $\lambda_0$ depends on $\tau$.
  4. In Figure 2 a specific choice of $J=\pi/4-0.05$ is made. Is there any particular reason for this? Perhaps a clarifying sentence.

---

## Round 3 · Referee Report · Pieter W. Claeys · 2023-6-14

Report

I would like to thank the authors for their detailed response. All my questions and comments have been appropriately addressed and I am happy to recommend this paper for publication in SciPost Physics.

---

## Round 3 · Author Response

Dear Editor,

we thank the Referees for carefully evaluating our manuscript, for their positive assesment, and their suggestions. We comment on their remarks in detail below.

Reply to Referee 1, Pieter Claeys:

1) The Referee points out an inconsistency of Eq. (21) and asks for the connections with the derivation cited in Ref. [65].

The permutation defined in (new) Eq. (21) differs from the cited reference for the following reason. The impurity interaction acts in the second layer of the circuit which generates the evolution of states in the Schrödinger picture whereas the folded impurity acts in the first layer of the super-circuit generating the evolution of operators in the Heisenberg picture. This leads to slight differences in the definition of the permutations defined in Eq. (21) and (64), respectively. In particular Eq. (21) does not correspond to Ref. [65] where the evolution of operators is described. We added a corresponding clarifying comment below the respective equations and moreover provide a simpler definition of the permutations $\sigma_\delta$.

2) The Referee asks for the entanglement dynamics, when the impurity interaction both supports a vacuum state and is T-dual.

In the minimal case of qubits demanding both T-duality and the existence of a vacuum state forces the impurity interaction to be of the form $\ket{0}\bra{0} \otimes v + \ket{0}\bra{0} \otimes w$ with diagonal single-qubit gate v and arbitrary single-qubit gate w (or with the two qubits swapped). This does not lead to entangling dynamics for the initial product states considered. For larger local Hilbert space dimension there are examples leading to non-trivial entanglement dynamics. A simple example is given by folded T-dual gates, for which the vectorized identity deals as a vacuum state. This however, is a special example, as in this case the vacuum state is also conserved upon evolution in "spatial" direction (of the gates entering the construction of the transfer matrices). Whether there are T-dual gates (for $q>2$) with vacuum state, which do not originate from such a folding procedure and which give rise to non-trivial entanglement dynamics is an open question. We now mention this after introducing T-dual gates, i.e., below Eq. (43).

3) The Referee criticizes that the discussion of the entanglement dynamics of states from generic impurity interactions is too brief.

We expanded the discussion of the entanglement dynamics for this situation, including a more detailed description of the origin of the plateaus and the subleading contributions as well as comments on different R\'enyi orders $n$. We find qualitatively similar behavior as shown in Fig. 3 with positive probability, when sampling the impurity interaction Haar random. Similarly, we also find gates, for which the subleading contribution is essentially irrelevant, with (larger) positive probability, which we now also mention in the text

Moreover, the Referee asks for the fate of the exact results upon perturbing away from T-duality or the existence of a vacuum state.

For the latter case one can obtain a rough picture as follows: For an impurity which supports an vacuum state the leading part of the transfer matrices' spectra lead to the reduced density matrix being a pure state. Perturbing away from such gates with vacuum states will cause the contribution from the leading eigenvalue to slightly differ from a pure states and hence yields non-zero entanglement thereby setting the value of $R_n$ on the plateaus. The subleading contributions, which encode the full entanglement dynamics in the unperturbed case (gates with vacuum), still give an important contribution upon small perturbation. However, they now add to non-zero entanglement entropy of the leading contribution ('on top of the plateaus') instead of the zero leading entanglement entropy in the unperturbed case. In this sense, the entanglement entropies depicted in Fig. 3, in which subleading contributions are clearly visible, indicates some proximity of the chosen gate to gates with vacuum state. We incorporated this picture into the text, when explaining the observed entanglement dynamics from generic impurities qualitatively.

In contrast, the lack of control over the subleading part of the spectrum and the associated eigenvectors of transfer matrices does not allow for a similar picture for perturbations of T-dual gates both for the entanglement dynamics of states and operators.

Additionally, as suggested by the Referee, we added Appendix C, in which we list all the impurity interactions used for numerical computations.

4) The Referee points out that the transfer matrices in the tensor network (25) correspond to the rows rather than the columns.

We changed the description of the network accordingly.

5) The Referee asks about numerical results for R\'enyi entropies of higher order $n>2$.

We have checked different orders $n$ for the R\'enyi entropies numerically. For the case of gates with vacuum and states as well as generic gates and operators, the entanglement due to the leading eigenvalues of transfer matrices is that of a pure state and hence independent from $n$. Similarly the exponential scaling with $\delta$ of subleading terms is independent from $n$. The prefactor (which we cannot compute explicitly) does weakly depend on $n$. For both operators and states in the T-dual case as well as for states in the generic case we numerically find almost no differences for different $n$. In particular the value of the plateaus for fixed $\tau>0$ does depend only very weekly on $n$. Subleading contributions are qualitatively similar for different $n$ but our analytical methods do not allow for quantifying the subleading contributions. We now comment on this when discussing numerical results, i.e., below Eq. (41) and (61) as well as in the extended Sec. 3.2.3. for states and below Eq. (69) and (109) for operators.

6) We corrected the typos pointed out by the Referee.

Reply to Referee 2:

1-3) As suggested by the Referee, we corrected the typos pointed out by the Referee, stated the values for $L$, $l$ and $t$ below Eq. (25), and explicitly mention the $\tau$ dependence of the subleading eigenvalue $\lambda_0$ below Eq. (40).

4) The Referee asks for the reasons to choose the interaction parameter $J=\pi/4 - 0.05$, when discussing T-dual gates.

This is done to ensure that the system is chaotic. In fact the point $J=\pi/4$ for qubits corresponds to the most chaotic gates (in the sense of Ref [73]). We choose $J$ slightly away from this point, to be in a more generic, but still chaotic regime. We now state this below Eq. (42) and in the caption of Fig. 2.

---

## Round 3 · List of Changes

1) We reformulated Eqs. (21) and (64) and comment on the origin of their differences.

2) We add discussion on impurity interactions which are both T-dual and support a vacuum state below Eq. (43)

3) We expand the discussion in Sec. 3.2.3. on the entanglement dynamics for generic impurity interactions.

4) We changed 'columns' to 'rows' in the description of the network (25) and added the concrete values of $L$, $l$, and $t$.

5) We added comments on numerical results for R\'enyi entropies of order $n>2$ below Eq. (41) and (61) as well as in the extended Sec. 3.2.3. for states and below Eq. (69) and (109) for operators

6) We explicitly mention the $\tau$ dependence of subleading eigenvalues of transfer matrices below Eq. (40)

7) We comment on the choice of the parameter $J$ in Eq. (42) below the equation and in the caption of Fig. 2

8) We corrected the typos pointed out by the Referees

9) We added Appendix C, in which we list all impurity interactions used for numerical computations.

You are currently on this page

Resubmission 2301.08168v3 on 5 June 2023

---

## Editorial Decision

published